

# Unexpectedly acidic nanoparticles formed in dimethylamine-ammonia-sulfuric acid nucleation experiments at CLOUD

Michael J. Lawler[1,2,a], Paul M. Winkler[3], Jaeseok Kim[4,5], Lars Ahlm[6], Jasmin Tröstl[7], Arnaud P. Praplan[8,9], Siegfried Schobesberger[8,10], Andreas Kürten[11], Jasper Kirkby[11,12], Federico Bianchi[8], Jonathan Duplissy[8], Armin Hansel[13], Tuija Jokinen[8], Helmi Keskinen[4,8], Katrianne Lehtipalo[8,7], Markus Leiminger[13], Tuukka Petäjä[8], Matti Rissanen[8], Linda Rondo[11], Mario Simon[11], Mikko Sipilä[8], Christina Williamson[11,14], Daniela Wimmer[8,11], Ilona Riipinen[6], Annele Virtanen[3], and James N. Smith[1,a].

[1]University of California, Irvine, Department of Chemistry, Irvine, CA, 92697, USA
[2]visitor at National Center for Atmospheric Research, Atmospheric Chemistry Observations and Modeling Lab, Boulder, CO, 80301, USA
[3]University of Vienna, Faculty of Physics, 1090 Vienna, Austria
[4]University of Eastern Finland, Department of Applied Physics, Kuopio, Finland
[5]Arctic Research Center, Korea Polar Research Institute, Yeonsu-gu, Incheon 21990, Republic of Korea
[6]Stockholm University, Department of Environmental Science and Analytical Chemistry Stockholm, Sweden
[7]Paul Scherrer Institute, Villigen, Switzerland
[8]Department of Physics, University of Helsinki, FI-00014 Helsinki, Finland
[9]Finnish Meteorological Institute, 00101 Helsinki, Finland
[10]Department of Atmospheric Sciences, University of Washington, Seattle, WA 98195, USA
[11]Goethe-University of Frankfurt, Institute for Atmospheric and Environmental Sciences, 60438 Frankfurt am Main, Germany
[12]European Organization for Nuclear Research (CERN), Geneva, Switzerland
[13]University of Innsbruck, Institute for Ion and Applied Physics, 6020 Innsbruck Austria
[14]Cooperative Institute for Research in Environmental Sciences, University of Colorado Boulder, Boulder, CO, USA and Chemical Sciences Division NOAA Earth System Research Laboratory, Boulder, CO, USA

[a]formerly at University of Eastern Finland, Department of Applied Physics, Kuopio, Finland

*Correspondence to*: Michael J. Lawler (mlawler@uci.edu)

**Abstract.** New particle formation driven by acid-base chemistry was initiated in the CLOUD chamber at CERN by introducing atmospherically relevant levels of gas phase sulfuric acid and dimethylamine (DMA). Ammonia was also present in the chamber as a gas-phase contaminant from earlier experiments. The composition of particles with volume median diameters (VMDs) as small as 10 nm was measured by the Thermal Desorption Chemical Ionization Mass Spectrometer (TDCIMS). Particulate ammonium-to-dimethylaminium ratios were higher than the gas phase ammonia-to-DMA ratios, suggesting preferential uptake of ammonia over DMA for the collected 10-30 nm VMD particles. This behavior is not consistent with present nanoparticle physico-chemical models, which predict a higher dimethylaminium fraction when $NH_3$ and DMA are present at similar gas phase concentrations. Despite the presence in the gas phase of at least 100 times higher base concentrations than sulfuric acid, the recently formed particles always had measured base:acid ratios lower than 1:1. The lowest base fractions were found in particles below 15 nm VMD, with a strong size-dependent composition gradient that suggests a change to a mixed-phase state as the particles grew beyond this size. The reasons for the very acidic





composition remain uncertain, but a possible explanation is that the particles did not reach thermodynamic equilibrium with respect to the bases due to rapid heterogeneous conversion of SO$_2$ to sulfate. These results indicate that sulfuric acid does not require stabilization by ammonium or dimethylaminium as acid-base pairs in particles as small as 10 nm.

# 1 Introduction

Atmospheric new particle formation (NPF) refers to the formation and subsequent growth of condensed phase particles from gas phase precursors. The mechanisms and chemical species responsible for NPF have been the subject of many recent studies, with the motivation of understanding the impacts of these processes on air quality and climate. NPF is an important source of cloud condensation nuclei (CCN) and therefore may significantly affect the global radiative energy balance (Wang and Penner 2009; Kazil et al., 2010), but the mechanisms and chemical species driving NPF are still poorly understood (e.g.

Riipinen et al., 2012). Sulfuric acid is widely accepted as one of the most important species for particle formation in the atmosphere (e.g. Kirkby et al., 2011; Kuang et al., 2008; Kulmala et al., 2007; Weber et al. 1997). However, particle growth rate and composition measurements have indicated that species other than sulfuric acid dominate growth past the smallest particle sizes in many environments (O'Dowd et al., 2002; Weber et al., 1997; Smith et al., 2008; Kuang et al., 2010; Bzdek at al., 2012). Three fundamental types of growth pathways for nanoparticles have been identified in the literature: reversible

condensational growth by low volatility species, reactive uptake, and acid-base interactions (Riipinen et al., 2012; and references therein). Recent evidence has supported an important role for highly oxidized, very low volatility organic molecules in the growth of atmospheric nanoparticles (Ehn et al., 2014; Zhao et al., 2013; Riccobono et al., 2014). Reactive uptake involves the multiphase transformation of a gas phase species into a new condensed phase species that is less volatile than its precursor (e.g. Wang et al., 2010). Growth due to acid-base chemistry results from strong ionic interactions, which

could be reversed depending on the chemical conditions within the cluster or particle (Bzdek et al., 2010; Barsanti et al., 2009).

    In the sulfuric acid-dimethylamine-ammonia-water system, the most likely growth pathways involve electrostatic interactions arising from proton transfer from sulfuric acid to the base species, direct condensational growth by sulfuric acid

due to its very low saturation vapor pressure, and coagulation of particles and/or clusters. Dimethylamine (DMA) and ammonia on their own are quite volatile gases, and significant growth by water vapor condensation alone is unfavorable due to the Kelvin effect until the particles are larger than about 50 nm in diameter. Nonetheless, all of these compounds may be incorporated together in a stable matrix as the particles grow. Measurements made at the CLOUD chamber in CERN of molecular clusters in the sulfuric acid-base system show a nucleation process involving stepwise additions of H$_2$SO$_4$ and

base, typically in a 1:1 ratio (Kirkby et al., 2011; Schobesberger et al., 2013; Almeida et al., 2013; Kürten et al., 2014; Bianchi et al., 2014; Schobesberger et al., 2015). At this early stage in the particle formation process, when there are fewer



than about 20 molecules in the clusters, ionic acid-base interactions clearly drive the growth. For these experiments, sulfuric acid was present at much lower concentrations than were the bases, so it was the chemical species that limited the growth.

Here we present chemical composition measurements of recently formed nanoparticles in the size range of 10-30 nm in diameter (volume median diameter for collected particles), from experiments conducted in the CLOUD chamber at CERN. These are the first direct compositional measurements of newly formed particles in this chemical system. These observations place constraints on the nature of particle growth after nucleation in the DMA-$NH_3$-$H_2SO_4$-water system, with implications for new particle formation in the atmosphere.

## 2 Laboratory Conditions

The experiments were undertaken at the Seventh Cosmics Leaving Outdoor Droplets (CLOUD7) campaign at CERN during October 2012. The CLOUD chamber is a 26.1 $m^3$ cylindrical chamber made from electropolished stainless steel. Efforts were taken in its construction to make it the cleanest possible chamber used for studying atmospheric particle nucleation (Kirkby et al., 2011; Schnitzhofer et al., 2014; Duplissy et al., 2016). It can be precisely temperature controlled, and UV light at 250-400 nm can be provided to the chamber with minimal heating via an array of fiber-optic bundles (Kupc et al., 2011).

It is possible to eliminate ions from the chamber using a 20 kV $m^{-1}$ clearing field, and conversely to increase the number of chamber ions by directing a 3.5 GeV pion beam into the chamber. The latter state results in ion concentrations that are up to an order of magnitude greater than what is present naturally from cosmic ray bombardment at the Earth's surface (Franchin et al., 2015).

Clean air and reagents were added to the chamber on a continuous basis, yielding a roughly 4-hour overturning time for the chamber. The main constituent gases were $N_2$ (79%) and $O_2$ (21%) from cryogenic liquid sources. The chamber was maintained at 38% relative humidity with ultrapure water generated by a Millipore filter system with UV irradiation (Merck Millipore). The trace gases added were ozone ($O_3$), dimethylamine (DMA), and sulfur dioxide ($SO_2$). $O_3$ was usually kept at 23.5 parts per billion by volume (ppb), and DMA was set to either 10 or 40 parts per trillion by volume (ppt) (see Table 1).

$SO_2$ was set to 63 ppb to facilitate the formation of $H_2SO_4$ at atmospherically relevant rates, leading to levels from 5 to 35 $\times10^6$ $cm^{-3}$. $SO_2$ was the only species added at levels well above likely background atmospheric mixing ratios. The addition of UV light resulted in OH radical formation, causing $SO_2$ oxidation to $H_2SO_4$ and formation of new particles. During CLOUD7 the chamber air was maintained at 278 K. Aerosol clearning cycles were performed regularly between experiments. For these cleanings, the chamber was flushed with clean humidified air and the particles were ionized with a

pion beam generated from the CERN Proton Synchrotron, with the high voltage clearing field was cycled on and off to allow charging and then removal of charged particles and clusters. An extended cleaning of the chamber was performed prior to the campaign. This involved rinsing the chamber walls with ultrapure water for 24 hours followed by raising the temperature





to 373K for a further 24 hours and flushing with ultrapure synthetic air to drive contaminants off the chamber walls. A further overnight heating cycle at 373K was performed prior to the experiment with no added DMA (the final experiment described here).

Despite the efforts to minimize contamination, ammonia was present as an impurity during the DMA experiments as a carryover from previous experiments during the campaign. A few days prior to the DMA-sulfuric acid experiments described in this study, nucleation experiments with intentional addition of ammonia gas at levels of 20 and 40 ppt were performed. After that time, no ammonia was intentionally added. However, ammonia was detected using ion chromatography (IC), as described in Praplan et al. (2012). The IC was not working during every experiment described here,

but it provided measurements of both $NH_3$ and DMA under several important experimental conditions. DMA was measured at an average mixing ratio of $46 \pm 12$ ppt (1 std dev) for nominal 40 ppt DMA experiments, for which a total of 18 particle collections are reported here, and $23 \pm 7$ ppt for nominal 10 ppt DMA experiments, for which a total of 21 particle collections are reported here (Simon et al., 2015). Over the respective periods, the average $NH_3$ was $28 \pm 9$ ppt and $19 \pm 10$ ppt. Therefore in both the high and low DMA cases, the DMA and $NH_3$ mixing ratios were comparable. It is worth noting

that the IC is likely to measure particulate ammonium and dimethylaminium as well as gas phase ammonia and dimethylamine. During the high temperature cleaning, ammonia reached extremely high levels (over 1000 ppt) due to evaporation from the chamber walls, whereas DMA only reached about 30 ppt. After the cleaning, both species were below the 5 ppt detection limit for the experiment with no added DMA.

Many instruments sampled the chamber air to assess the characteristics of the gases, molecular clusters, and particles present in the chamber. These included a Scanning Mobility Particle Sizer (SMPS), a Hygroscopicity Tandem Differential Mobility Analyzer (HTDMA, Kim et al., 2016; Keskinen et al., 2013), and several mass spectrometers. Atmospheric Pressure interface Time-of-Flight Mass Spectrometers (APi-TOFs, Tofwerk AG) measured gas phase ions (Kirkby et al., 2011; Schobesberger et al., 2013; Almeida et al., 2013; Bianchi et al., 2014), and two Chemical Ionization APi-TOFs measured

neutral gas phase species (Jokinen et al., 2012; Kürten et al., 2014).

### 3 TDCIMS Instrument

The Thermal Desorption Chemical Ionization Mass Spectrometer (TDCIMS) has been described in detail elsewhere (Smith et al., 2004; Lawler et al., 2014). Significant improvements in ammonium sensitivity were made since the measurements described in Lawler et al. (2014) by modifying the differential pumping and ion transfer optics. For these experiments,

particles were sampled continuously from the CLOUD chamber at 3.3 l min$^{-1}$ (Figure 1). Sampled particles were charged with a pair of unipolar chargers (UPCs, McMurry et al., 2009; Chen and Pui, 1999). The charged particles were then collected by electrostatic deposition onto a Pt filament. After a collection period of typically 30 minutes, the filament was



translated into the ion source of the mass spectrometer and the temperature was ramped to ~600 °C to desorb the collected mass. Desorbed molecules and decomposition products were ionized and ultimately detected using a high resolution time-of-flight mass spectrometer (HTOF, Tofwerk AG). Instrument backgrounds were assessed for each collection by performing the identical procedure but without applying a collection high voltage. All measurements reported here are corrected for this

background. The TDCIMS is capable of detecting both positive and negative ions, but only one polarity can be monitored for each sample collected. Chemical ionization reagent ions are generated by an $^{241}$Am source kept in a clean $N_2$ flow. Impurity $H_2O$ and $O_2$ in the $N_2$ gas result in $(H_2O)_nH^+$ reagent ions for positive mode, and $(H_2O)_nO_2^-$ reagent ions in negative mode. Particle phase dimethylaminium ($C_2H_8N^+$, or $DMAH^+$) and ammonium ($NH_4^+$) are detected as molecular ions and as ions clustered with water in positive mode. Sulfate is detected in negative ion mode primarily as the $SO_5^-$ ion, which is

formed from the reaction of the sulfate salt decomposition product $SO_3$ with $O_2^-$. The TDCIMS observations are presented as sums of background-corrected detected ions, integrated over the desorption period, for each collected sample, or as ratios of these ion sums.

During most of the CLOUD7 experiments, the sampled particles were not mobility-selected, and the particle sizes that made

up most of the collected mass were dependent on the chamber particle size distribution as it evolved over the course of the particle formation events. For one experiment, a mobility size of 20 nm was selected using a pair of radial differential mobility analyzers (RDMAs, Zhang et al., 1995) placed between the unipolar chargers and the electrostatic precipitator. This mostly served to exclude small (< 15 nm) particles from analysis. A nano-SMPS sampled particles after charging in the UPCs. The nano-SMPS consisted of a differential mobility analyzer (TSI Inc., model 3085; Nano DMA) and an ultrafine

condensation particle counter (TSI Inc., model 3027) and was capable of measuring particles of 4-42 nm mobility diameter. The nano-SMPS system was temporarily moved downstream of the electrostatic precipitator to assess the size-resolved efficiency with which the precipitator captured sampled particles. Additionally, a post-campaign test was performed using a long-column differential mobility analyzer (TSI Inc., model 3081; Long DMA) to assess the capture efficiency of 12-140 nm particles under the same instrumental operating conditions. The region of overlap between these two collection efficiency

measurements compared well. The two efficiency measurements were combined for correcting the experimental data, and the absolute cutoff for collection was determined to be about 60 nm (Figure 2). Charged particles smaller than about 15 nm appeared to be captured with ~100% efficiency once inside the precipitator region.

To assess the net size-resolved collection efficiency of the instrument, sampling losses and (multiple) charging efficiency

had to be taken into account. This was achieved by a statistical optimization procedure that relied on the CLOUD chamber SMPS system and a CPC downstream of the TDCIMS precipitator, using a procedure similar to that described in Lawler et al. (2014). Briefly, a size-resolved sampling and charging efficiency filter was optimized to obtain agreement of the chamber particle distribution with the TDCIMS CPC when there was no precipitator voltage applied and after correcting for UPC-generated particles (see next paragraph). This information was combined with the precipitator capture efficiency to





determine the sizes and total volume of particles collected. The volume median diameter (VMD) was assessed as a measure of the center of the collected particle distributions.

When sampling chamber air with the TDCIMS, there was a persistent mode of small (~ 6 nm) particles generated in the UPCs. This was a result of ionization and oxidation processes driven by the UPC radioactive sources. The particles were present even in the absence of $H_2SO_4$ in the chamber, indicating that oxidation of $SO_2$ to form $H_2SO_4$ was probably involved. The alpha-emitting radioactive sources are known to produce OH radicals from $H_2O$ present in the sampled air (e.g., He and Hopke, 1995). When these charger-produced particles were the only source of particles to the TDCIMS, the resultant
background-corrected particle phase ion signals were usually not detectable, indicating that these fine particles did not contribute significantly to the TDCIMS chamber measurements. However, an attempt was made to try to reduce any possible impact of this oxidation process, as it also seemed possible that charger-generated $H_2SO_4$ could influence the composition of sampled particles. During many of the experiments, a flow of 5% $H_2$ in Ar (industrial-grade welding gas) was added to the TDCIMS sampling inlet to achieve a ~1% $H_2$ mixing ratio. This was done to scavenge OH and other oxidants generated in
the UPCs and resulted in a reduction in the spurious small particle mode by a factor of 3-4. Measured particle sulfate fractions did not appear to be substantially affected by this change (see Results). However, this test demonstrated that n-methyl formamide detected in the particles was the result of oxidation processes within the charger (see Results). Adding the $H_2$ gas resulted in higher background signals and variability for many of the signals, due to contaminants in the $H_2$/Ar gas cylinder.

**3.1 Instrument Calibrations**

**3.1.1 Calibrations during the CLOUD7 measurements**

The TDCIMS was calibrated for ammonium sulfate (AS) particles once during the campaign by using an atomizer to generate aerosol from a dilute ammonium sulfate solution. The wet particles were passed through a diffusion dryer before entering the UPCs. This calibration aerosol was mobility-selected to either 15 or 20 nm using a pair of RDMAs, one
downstream of each UPC, to control the amount of mass available for collection on the filament. By selecting for 15 nm particles and collecting for 5 minutes, about 0.3 x$10^{-9}$ cm$^3$ of particle volume was collected. This calibration showed $SO_5^-$ and $(H_2O)NH_4^+$ signals in the low-to-mid range of those observed in the nucleation experiments (Figures 3 and 4). By selecting 20 nm particles, an estimated particle volume of 1.3 x$10^{-9}$ cm$^3$ was collected, resulting in sulfate signals that were similar to mid-to-large chamber aerosol samples and ammonia signals over twice as large as any observed for chamber
aerosol (Figures 3 and 4). These AS aerosol calibrations indicate that the TDCIMS was roughly equally sensitive on a molar basis to ammonium and sulfate during CLOUD7, and showed that the instrument had a roughly linear base:acid ratio over a range of collected masses (Figure 5).





### 3.1.2 Post-CLOUD7 Calibrations

Aerosol calibrations were performed after the CLOUD7 experiments to assess the relative sensitivity of the TDCIMS to particle ammonium and dimethylaminium and to assess the linearity of the response. Aqueous solutions of 2 mM ammonium sulfate and dimethylaminium bisulfate were atomized in a stainless steel atomizer, dried by passing through a thermal

denuder at 373 K, and sampled by TDCIMS. All transport, sheath, and atomizer gas flows used $N_2$ gas delivered from a liquid $N_2$ dewar. Tests in our lab have shown that even brief exposure to lab air or even "clean" zero air can cause significant contamination of the particles with ammonium. The ammonium sulfate solution was generated using ammonium sulfate salt (>99%, Sigma-Aldrich) and deionizied water (Millipore). The dimethylamine bisulfate solution was made using 40% (by mass) dimethylamine in $H_2O$ (Sigma-Aldrich), $H_2SO_4$ (95-98%, Sigma-Aldrich), and deionized water (Millipore) to form a

1:1 molar ratio of dimethylaminium to sulfate. The TDCIMS RDMAs were used to size-select the sampled particles at 25 nm mobility diameter, and a nano-SMPS was used to characterize the resulting size distribution. During particle collection, the nano-differential mobility analyzer was bypassed so that the number concentration difference between sample and background could be used to assess the number of particles collected for brief collection periods. Sample time was varied to achieve different collection masses, which were estimated using the known size distribution, the number concentrations

downstream of the wire during collections, and the wire collection efficiency, which was determined using a comparison of back-to-back stable collection and background particle concentrations. For these calibrations, the dry particle densities were assumed to correspond to dimethylaminesulfate (1.35 g cm$^{-3}$) and AS (1.78 g cm$^{-3}$), respectively (Qiu and Zhang, 2012), and collected masses are reported. It should be noted that the actual base:acid ratio of the calibration aerosol is not known exactly because of loss to the gas phase during evaporation. For example, Wong et al. (2015) found that their wet AS aerosol had an

ammonium:sulfate ratio of 1.72 due to this process. In addition, a given salt may only have certain stable configurations when dried. A dimethylamine sulfate droplet was shown to go from a 2:1 solution to a 1.5:1 DMAH$^+$:SO$_4^{2-}$ ratio particle when dried (Chan and Chan, 2013). Ouyang et al. (2015) showed a tendency for dried DMA:$H_2SO_4$ nanoclusters to reach a consistent density of 1.567 g cm$^{-3}$, independent of the bulk composition of the initial electrospray solution. Despite using solution DMA:$H_2SO_4$ molar ratios ranging from 1:10 to 2:1, the particles settled on the same final composition. Similarly,

during "droplet tests" in our lab in which dilute salt solutions are directly applied to the TDCIMS collection wire, allowed to evaporate, then analyzed normally, we found only a few percent increase in DMAH$^+$ signal when using a 2:1 DMAH$^+$:SO$_4^{2-}$ solution compared with a 1:1 solution, indicating that the additional DMA simply evaporated rather than being incorporated into the salt (the molar ratio of which remains uncertain).

The sensitivity of the TDCIMS to sulfate in the negative ion mode was basically indistinguishable for the two salt types (Figure 6). The signal dependence was nonlinear with respect to sampled mass and the relationship is well-fit by a second-order polynomial. Similar to the negative ion signal, the base signals showed nonlinear sensitivity with respect to mass when a large range of collected mass was considered, and the relationship could be described using a second-order polynomial.





$NH_4^+$ was detected as a contaminant in the dimethylaminium bisulfate aerosols. Contamination of the particles by ammonia most likely occurred sometime prior to collection on the wire (i.e. either during generation or transport of the particles), based on tests of temporal stability of the particle composition after precipitation onto the collection wire. The solutions themselves showed very little ammonium contamination, based on a test in which a dilute solution was directly applied to

the precipitator wire. This test also indicated that there was no significant ammonia released upon thermal decomposition of the dimethylaminium salt. These observations underscore the facility with which ammonia is taken up into DMA-$H_2SO_4$ aerosol. The $DMAH^+$ signals for the DMABS calibration particles were slightly less than but comparable to the $NH_4^+$ signals for the AS calibration particles. If the DMABS calibration aerosol were exactly 1:1 $DMAH^+$:$SO_4^{2-}$ and the AS aerosol were exactly 2:1 $NH_4^+$:$SO_4^{2-}$, this would indicate higher sensitivity to $DMAH^+$ than to $NH_4^+$ such that the reported CLOUD7

experimental $DMAH^+$:$NH_4^+$ ratios are overestimates. In this worst case, the $DMAH^+$ sensitivity is about 50% higher than the $NH_4^+$ sensitivity. However, based on the discussion presented above, the actual composition ratios were probably closer for the two calibration aerosols, and any inferred sensitivity difference would be smaller. If the actual composition ratios for the AS and DMABS aerosol were 1.75:1 $NH_4^+$:$SO_4^{2-}$ and 1.25:1 $DMAH^+$:$SO_4^{2-}$, respectively, the inferred sensitivity to both bases would be the same. The base:acid signal ratios were roughly linear over a wide range of collected masses (Figure 7),

allowing a straightforward analysis of this quantity for the chamber aerosol. Overall, these calibrations indicate no significant sensitivity bias for the salts studied, but there is uncertainty in the relative sensitivity of $DMAH^+$ and $NH_4^+$ due to uncertainty about the calibration aerosol composition.

An additional test was performed after the CLOUD7 experiments to determine the stability of DMA-$H_2SO_4$ nanoparticles on

the TDCIMS filament. Nanoparticles were formed using high concentrations of DMA and $H_2SO_4$ in a glass flow reactor modeled after that described in Glasoe et al. (2015). Particles of ~5-50 nm were generated and sampled with the TDCIMS. Particles were analyzed both in the usual way, with desorption proceeding a few seconds after the conclusion of sampling, and with an additional 'rest' period three times as long as the sampling period before desorption to allow any volatile compounds to desorb. There was no significant difference in the results from the two methods, indicating that, for these

particles at least, continuous evaporation of base was not an issue. This test does not exclude the possibility that some DMA is lost rapidly during sampling, i.e. as the result of the evaporation of an aqueous phase. However, the calibrations performed with atomized solutions indicate that $(DMA)H_2SO_4$ and $(NH_3)_2H_2SO_4$ aerosol behave similarly in this regard.

### 3.1.3 Overview of calibrations

To summarize the calibration results, the TDCIMS shows identical sensitivity to the sulfate fraction of the two calibration

aerosols tested, DMABS and AS. On the other hand, there is some uncertainty in the relative sensitivity of the basic components $NH_4^+$ and $DMAH^+$. If there is indeed any difference, the instrument is more sensitive to $DMAH^+$ and therefore may overestimate both $DMAH^+$:$NH_4^+$ and base:acid ratios. However, the measured experimental ratios were in fact lower than expected (see Results). The ammonium sulfate calibration performed at CLOUD7 should be considered the best





assessment of the relative sensitivities between positive (base ions) and negative (sulfate ions) ion modes for these measurements because it was performed under identical operating conditions. This calibration showed the TDCIMS to be slightly more sensitive towards particulate ammonium than towards particulate sulfate, but given the very low base:acid ratios measured for chamber aerosol (see Results below), no attempt was made to correct for this, and the base:acid ratios

presented are strictly the ratios of the uncorrected ion signals.

## 4 Results

During the CLOUD7 experiments, negative ion TDCIMS spectra were dominated by $SO_5^-$ (Figure 8a). This was also the strongest signal for ammonium sulfate in the calibrations (Figure 4a) and results from the reaction of $SO_3 + O_2^-$ in the ion source following the decomposition of sulfate salts. There were other sulfur masses that tracked $SO_5^-$, including $SO_3^-$, $SO_4^-$,

and $HSO_4^-$. The oxidized sulfur signal per estimated collected mass is consistent with the ammonium sulfate calibration aerosol, indicating that sulfate dominated the collected mass in the particles measured from the chamber, as expected (Figure 3). Overall the spectra were very clean, but some apparent contaminant organic molecules were also detected in the particles, including $C_3H_3O_3^-$ (an oxidized carboxylic acid or fragment), $CHNO_2^-$ (likely fragment ion), and $C_8H_4O_3^-$ (phthalic anhydride). Phthalic anhydride is most likely derived from phthalate contamination by plastics.

Dimethylaminium and ammonium were major components of the positive ion spectra (Figure 8b). The main ammonium peak was its first water cluster, $NH_4^+(H_2O)$, which is also the main ammonium peak in the ammonium sulfate standards (Figure 4b). There was also a large $C_2H_6NO^+$ peak, which appears to be the DMA oxidation product n-methyl formamide (MFA), a contaminant generated in the UPCs (see below and Figure 9). In all experiments, more ammonium was detected in

the particles than dimethylaminium and no clear difference between nominal 40 ppt and 10 ppt DMA runs was observed (Figure 10). The $DMAH^+:NH_4^+$ ratio was more likely to be lower for smaller amounts of collected mass (Figure 10), though a clear dependence on particle size was not observed. The collected mass dependence could be due to a limitation of the available contaminant ammonia or the aging time necessary to change the composition. Small mass collections were often the result of sampling at the end of runs, when the remaining particles were few, large, and aged, with no more sub-10 nm

particles present and collected VMDs closer to 30 nm. In the run conducted with no DMA added, the $DMAH^+$ signal was about 1% of the $NH_4^+$ signal, indicating that DMA did not remain as a significant contaminant in the chamber, unlike $NH_3$.

To investigate the apparent spurious n-methyl formamide (MFA, $C_2H_6NO^+$) formation in the UPCs, a test was performed in which the oxidant-scavenging $H_2$ flow to the TDCIMS inlet was turned off midway through a particle nucleation experiment.

This experimental condition was maintained for a period of over three hours during the nucleation experiment, such that two measurements under "high charger oxidant" conditions were made for both positive and negative ion modes. Most of the main species observed were only minimally affected by this test, but $C_2H_6NO^+$ in the particles increased by over an order of



magnitude (Figure 9). The DMAH$^+$ signal was not strongly affected, indicating that DMA present in the sampled particles was not appreciably oxidized in the absence of the H$_2$ scavenger. This suggests that the DMA oxidation to MFA occurred in the gas phase and the MFA partitioned to the particles afterward. It is also possible that the instrument sensitivity for MFA is much greater than for DMAH$^+$ and only a small fraction of particle phase DMAH$^+$ was oxidized. In either case, the large

MFA signal is clearly spurious and we have no evidence suggesting that the particle phase DMA signal was significantly affected by the MFA production. MFA signal was therefore excluded from calculations of base:acid and DMAH$^+$:NH$_4^+$ ratios. Another concern was that other species, especially H$_2$SO$_4$, could be generated in the chargers and partition to the particles. The addition of H$_2$ reduced the fraction of charger-generated OH that could react with SO$_2$ by almost two orders of magnitude. However, no consistent bias in base:acid ratios is evident when H$_2$ was added (see next paragraph, and Figure

11). Overall, therefore, there is nothing to indicate that the measured base:acid ratios were strongly biased by sampling artifacts.

Ratios of the sum of base (DMA and ammonia) to acid (sulfate) integrated peaks were calculated to assess the acidity of the particles in each experiment (Figure 11). Because acid and base compounds are each detected in a different polarity

(negative and positive, respectively), there is always a time gap between the observations, during which the particle size and potentially composition have changed. However, reasonably consistent results were achieved when comparisons were made for collections close in time for similar volume median diameters of sampled particles, with similar amounts of collected mass. There was not a strong relationship between gas phase sulfuric acid concentration and particle composition. Kim et al. (2016) similarly found a non-dependence of 10 nm particle hygroscopicity on sulfuric acid, but did observe a decrease in

base:acid ratio for 15 nm particles as gas phase sulfuric acid increased. For essentially all collections, the TDCIMS base:acid ratio was below 0.5, indicating that the particles did not reach a fully neutralized state. For some collections of small (~ 10 nm VMD) particles, this ratio was near 0.1, though there was high variability for this size range. The HTDMA data show a similar size dependence, but indicate higher base:acid ratios (Figure 11). A notable exception to the very acidic particles was a case for which particles were mobility-selected at 20 nm before collection and the base:acid ratio was 1.23. This is most

likely due to contamination of the particles by ammonia in the recirculating sheath flow of the mobility analyzers and so the results of this experiment are therefore excluded from Figures 10 and 11. The DMAH$^+$:NH$_4^+$ ratio was low (~ 0.15) for this experiment, indicating a substantial change in particle composition relative to other experiments. The mass of small particles excluded during this test was too small to account for the large change in the base:acid ratio, even if the excluded particles contained no DMA or ammonia. It should be noted that the same mobility analyzers were used to size-select the calibration

ammonium sulfate aerosol (see Calibrations section). However, it is unlikely that the calibration particles could become contaminated by ammonia to the extent that they achieve a base:acid ratio higher than that of the stable ammonium sulfate salt (i.e. 2:1), which is roughly the expected composition of the calibration aerosol. So, while the size-selection process significantly impacted the composition of the acidic chamber particles, this process was unlikely to influence the composition of the calibration aerosol greatly.





## 5 Discussion

### 5.1 Particle phase dimethylaminium-ammonium ratios

During these experiments, contaminant ammonia was present in the chamber at significant levels as a result of earlier experiments with intentional ammonia additions. TDCIMS measurements indicate that this ammonia was incorporated into growing sulfuric acid particles of ~10-30 nm with an efficiency comparable to or greater than DMA. Observations of nucleated clusters in the CLOUD chamber show that DMA is an important constituent of recently nucleated particles when it is present in the chamber at concentrations as small as 5 ppt (Almeida et al., 2013; Kürten et al., 2014), whereas ammonia was only detected in clusters with more than 7 $H_2SO_4$ molecules (Bianchi et al., 2014). However, it is possible that ammonia molecules were present in smaller clusters in the chamber and evaporated during analysis in the APiTOF of CI-APiTOF instruments. The abundance of ammonia in the TDCIMS-measured particles indicates either that the molecular clusters that lead to > 10 nm particles contain more ammonia than is thought, or that ammonia is incorporated into > 10 nm particles by a process distinct from cluster formation, or both.

A particle composition model based on chemical equilibria for aqueous well-mixed particles predicts roughly equal fractions of $DMAH^+$ and $NH4^+$ ions in the particles when 15 ppt ammonia and 10 ppt DMA are present in the gas phase (Ahlm et al., 2016). At smaller DMA concentrations ammonium is expected to dominate. For gas phase concentrations of 40 ppt DMA and 15 ppt ammonia, however, the particulate phase base should be primarily $DMAH^+$ with a $DMAH^+$: $NH_4^+$ ratio of about 5, due to the higher basicity and lower saturation vapor pressure of DMA (Ahlm et al., 2016). The relative $DMAH^+$: $NH_4^+$ ratio is thus expected to be highly dependent on the gas-phase concentration of DMA, a feature which is not observed in the TDCIMS data. This may be explained in part by the fact that the DMA:$NH_3$ ratio in the chamber, as measured by IC, was actually comparable in both the 40 ppt and 10 ppt DMA experiments. That is, the ammonia contamination was somewhat lower for the 10 ppt DMA experiments. However, for DMA:$NH_3$ gas phase ratios above 1, there should be more $DMAH^+$ than $NH_4^+$ in the particles. The particle phase $DMAH^+$: $NH_4^+$ ratios are close to 1 in many cases, such that a relatively small difference in TDCIMS sensitivity to the two bases, or the relatively large uncertainty in the gas phase DMA:$NH_3$ ratio, would be enough to remove disagreement from expectations. However there are several cases for which the $DMAH^+$ signal is constrained by error bars to be less than 2/3 of the $NH_4^+$ signal, and based on calibrations, the instrument sensitivity to $DMAH^+$ appears to be the same or higher than to $NH_4^+$. This unexpected result has at least two potential explanations: 1) a deficiency in the thermodynamic calculations of the particle phase composition, such as a kinetic limitation for surface uptake of the bases or water; or 2) a measurement error or artifact in the particle or gas-phase composition. In the following we discuss each of these possibilities.

Models that are based on chemical equilibria for well-mixed aqueous particles may not adequately describe nanoparticles. Recently-formed sulfuric acid nanoparticles may have structures and corresponding intermolecular interactions that are





different from those in a well-mixed solution. For example, steric effects that favor the presence of ammonia over DMA may be important at these small sizes. Thermodynamic calculations and previous experimental studies on larger particles suggest aqueous mixtures of aminium and ammonium sulfates with base:acid ratios from 2:1 to 1:1 are the stable composition at the RH of the CLOUD chamber experiments (Chan and Chan, 2013; Sauerwein et al., 2015). However, there were unfortunately

no direct observations on the phase and water content of the nanoparticles in these experiments.

Kinetic limitations to surface uptake of the bases are probably only possible for non-aqueous particles, since the diffusion time into an aqueous 10 nm particle is very short compared to the lifetime of a particle in the chamber. However, recent work on nanoparticle phase suggests that these small particles should be aqueous (Cheng et al., 2015). The growth rates of 2

nm particles in the CLOUD7 experiments are around ten times faster than would be predicted if the growth were due to perfectly efficient condensation of the measured gas phase sulfuric acid monomers (Lehtipalo et al., 2016). This suggests that cluster-particle collisions were the main driver of the growth of the newly formed sub-2 nm particles. The calculations presented by Ahlm et al. (2016) suggest that coagulation was also an important factor driving the growth of the larger particles. The composition of the smaller clusters and particles could therefore be reflected in the composition of the growing

particles, if the timescale of the growth was significantly shorter than the equilibration timescale of the particles with the gas-phase bases (such as would occur for high viscosity, non-aqueous particles). However, there is no evidence indicating an enhancement of $NH_4^+$ relative to $DMAH^+$ in smaller particles and clusters, and, according to present understanding, most particles should have reached thermodynamic equilibrium in the chamber before sampling.

Experimental errors that would affect these interpretations are limited to differential measurement sensitivity between $DMAH^+$ and $NH_4^+$ in the particle phase, and between DMA and $NH_3$ in the total (mostly gas phase) observations. In the case of the TDCIMS particle phase measurements, this could arise from a sampling artifact involving either preferential loss of DMA relative to ammonia or contamination of the sample with ammonia. DMA is expected to form a more stable salt with sulfuric acid than is ammonia, making the former possibility unlikely. Calibrations with known standards also show no

relative deficiency of the DMA signal (see Calibrations). Significant contamination of the particles by ammonia within the TDCIMS is also unlikely given the insensitivity of the measured base:acid ratios and $DMAH^+$:$NH4^+$ ratios to the ammonium background measured by the TDCIMS. Ammonium backgrounds were over an order of magnitude higher when $H_2$ gas was added, but the particle phase signals were not measurably influenced (Figures 10 and 11). Furthermore, when $H_2$ gas was not added to the TDCIMS inlet, most of the background ammonium signal came from the chamber. After the high temperature

chamber cleaning, which did not affect the TDCIMS (it was disconnected), the ammonium background went down by a factor of 5. This indicates that most of the ammonia that was available to be picked up by particles was provided by desorption from the chamber walls, not by contamination within the instrument. The gas phase DMA observations could have been overestimated with respect to ammonia if the DMA was preferentially in the particle phase and ammonia was not, and if the ion chromatography system was sensitive to both particle phase and gas phase DMA. The TDCIMS measurements





are not sensitive to gas phase compounds, however, and they indicate that there was not preferential particle uptake of DMA compared with ammonia. Nonetheless, for the larger mass collections, a significant fraction of both the DMA and $NH_3$ present in the chamber was present in the particle phase.

To summarize, the $DMAH^+:NH_4^+$ ratios in the particles appear to be lower than expected based on thermodynamic equilibrium calculations. However, there is significant uncertainty in the actual gas phase mixing ratios of DMA and ammonia in the chamber, and it is also possible that the TDCIMS showed differential sensitivity with respect to $DMAH^+$ and $NH_4^+$ in the salts that constituted the particles in the chamber. Given these uncertainties, we cannot exclude the possibility of no deviation in the $DMAH^+:NH_4^+$ ratios from thermodynamic expectations. However the observed base:acid ratios in the
particles were far from their expected values, as discussed below.

### 5.2 Particle phase base:acid ratios

The TDCIMS results indicate a base:acid ratio below 0.5 for particles 10 nm and above, despite a significant excess of gas
phase DMA and ammonia with respect to sulfuric acid. This is counter to the expectation that the particles should reach thermodynamic equilibrium and be closer to pH-neutral, with more ammonium and dimethylaminium ions than sulfate ions (Ahlm et al., 2016). This discrepancy may be explained by uncertainties in the thermodynamics of small particles, but oxidative chemistry may also play a role in this case. Possible experimental biases are discussed below as well.

As mentioned previously, CLOUD measurements and numerical calculations of cluster and nanoparticle growth show that cluster-cluster, cluster-particle, and particle-particle collisions are important drivers of growth in the DMA-sulfuric acid system (Lehtipalo et al. 2016, Ahlm et al. 2016). The composition of clusters could therefore be important for determining the composition of > 10 nm particles for timescales shorter than the equilibration timescales of the particles. Positive ion APi-TOF mass spectrometer measurements (not shown) indicate that up to the maximum observable sizes for this technique
(~ 2 nm), the composition of ammonia-DMA-sulfuric acid clusters may is acidic for the smallest clusters but approaches a 1.2:1 base:acid composition (Bianchi et al. 2014, Figure 11). It seems possible that the least-neutralized (most acidic) clusters could have the highest accommodation coefficients and therefore be the most important clusters for nanoparticle growth, but we could find no evidence to support this in the literature. This hypothesis could explain why the observed >5 nm particles are more acidic than most of the observed < 2 nm clusters. The most effective clusters may have shorter
lifetimes to accommodation and may therefore be less prominent in the APi-TOF spectra. It should be noted that equilibrium cluster distribution measured by the APi-TOF gives essentially no information about the relative importance of the different clusters for nanoparticle growth, due to a lack of knowledge about the cluster formation rates and lifetimes. In any case, it





appears clear that the growth of particles above 5 nm is not due to a stepwise addition of $H_2SO_4$ and DMA molecules as appears to be the case for the smallest clusters.

In view of the apparent production of $H_2SO_4$ in the UPCs, it is necessary to consider the possibility that the low base:acid

ratios resulted from contamination of the particles by sulfuric acid within the instrument. There are two distinct periods when such contamination could have occurred: 1) while the particles were within the unipolar chargers, and 2) while the particles rested on the precipitator wire after being collected. Significant growth in the first, brief period can be excluded by comparing the appearance times of particles of a given size in the chamber with those for the particles exiting the UPCs, as measured with the differential mobility analyzer and CPC downstream of the UPCs. There was no discernible growth during

this period. Addition of compounds to the precipitator wire should be inefficient due to the flow of $N_2$ sheath gas around it. This does nonetheless occur, but the effects of this are in general subtracted out by the background correction. A scenario in which this background correction would not be sufficient would be if the gas phase $H_2SO_4$ were more efficiently incorporated into sampled particles on the wire than onto a particle-free wire. Figure 3 suggests that this was not a significant issue, since the sulfate signal was consistent with the sampled particle mass. The estimation of collected particle mass cannot

be influenced by processes occurring after collection onto the wire.

The TDCIMS data overall indicate a large increase of base fraction as the particles grew from 10 to 20 nm (Figure 11), though individual experiments showed high variability for ~ 10 nm particle composition. The direction of this composition change is corroborated by aerosol hygroscopicity results from the same experiments. These measurements showed that 15

nm particles had significantly lower hygroscopicity than 10 nm particles (Kim et al., 2016). Kim et al. estimated a base:acid ratio of roughly 1.0 for 15 nm particles and 0.3 for 10 nm particles, based on the measured hygroscopicities. Our results indicate even lower ratios. Given the uncertainty of TDCIMS sensitivities to very acidic particles, the TDCIMS ion ratio results are estimates below a base:acid ratio of ~ 1.0. The differences in the absolute base:acid ratio between the TDCIMS and HTDMA based measurements could be related to the handling of the particles during sampling. In the case of the

TDCIMS observations, the sampled particles were exposed to clean $N_2$ gas for up to 30 minutes. This is unlikely to introduce much contamination but may result in the loss of molecules that do not form a stable salt while the particles dry in the $N_2$ flow. The final composition and phase of the drying particles is uncertain and probably depends on size. However, the many calibrations performed during and after CLOUD7 indicate that the TDCIMS analysis does not show a strong tendency to lose bases prior to analysis for ammonium or dimethylaminium sulfate aerosol (see Calibrations section). That the base:acid

ratio is not strongly biased within the instrument is shown by two observations of very different particle phase base:acid ratios from most of the DMA-$H_2SO_4$ experiments described here: (a) we measured calibration aerosol with much higher base:acid ratios than the aerosol formed in the experiments described here (see Calibrations), and (b) when we exposed the particles to some air of unknown composition by size-selecting them in one experiment, we contaminated the particles with ammonium and found the base:acid ratio to increase (see Results). The operation of the HTDMA instrument depends on the



addition of humidified and dried HEPA-filtered air. It seems likely that these observations of base:acid ratio could be biased higher than the TDCIMS observations due to contamination by ammonia. In any case, the independent observations of particle chemistry and physical properties provided by the TDCIMS and HTDMA agree that particles at 10-20 nm sizes had a substantially larger base ion fraction than particles near 10 nm.

The large change in particle composition around 10 nm indicates a sharp change in particle compositional growth. This could result from a phase change from a purely liquid state to a mixed phase state with a solid core. Recent results have shown that the phase state of nanoparticles is size-dependent, such that particles are expected to be completely liquid below some size threshold, even for supersaturated droplets (Cheng et al., 2015). The location of this transition depends on particle

size, composition, water content, and temperature, and it was demonstrated to occur for ammonium sulfate nanoparticles at around 10 nm at 298 K for a particulate ammonium sulfate mass fraction of 0.63 (Cheng et al., 2015). The molar concentrations of the species present in the smaller, purely liquid particles are much higher than those in the liquid phase of the slightly larger mixed-phase particles. This could explain the strong difference between 10 nm and larger particles, through a few possible mechanisms. There may be an enhancement of interfacial and/or aqueous phase reaction rates for the

smaller particles relative to the larger, phase-separated particles. In addition, base molecules present in the aqueous phase are more likely to be lost as the particles evaporate during sampling than base molecules held in salts in a solid phase. Therefore, larger phase-separated particles may be better able to maintain their base molecules when sampled by the TDCIMS. However, the HTDMA measurements also show a lower base:acid ratio in the 10 nm particles, and sampled particles in this instrument do not experience a long period in dry, clean air during which base molecules can slowly evaporate. Overall, the

evidence suggests that the apparent high acidity of the small particles is a feature of the chamber particles and not an artifact of the experimental methods.

We can also consider whether the high chamber $SO_2$ mixing ratios could have led to low base:acid ratios in the particles. $SO_2$, or S(IV), can be oxidized to sulfate in aqueous droplets by $H_2O_2$ and $O_3$ (e.g. Caffrey et al., 2001; Hoyle et al., 2016).

We are aware of only one study examining the potential for known aqueous phase S(IV) oxidation reactions to cause significant particle growth in nanoparticles (Kerminen et al., 2000). This numerical study showed that known pathways cannot result in enough growth for newly formed nanoparticles to reach CCN size, given typical tropospheric particle lifetimes and oxidant and $SO_2$ mixing ratios of 100 ppt. However, the $SO_2$ mixing ratio in the present experiments was 63 ppb, typical of pollution plumes, and there was no larger aerosol mode to act as a coagulation sink for the small particles and

thereby limit their growth. Furthermore, $H_2O_2$ levels were also probably higher than typical ambient levels, as this species is the main $HO_2$ radical sink under the CLOUD chamber conditions of extremely low NOx. In addition to this well-known chemistry, a recent study has demonstrated an extremely rapid surface reaction of $SO_2$ on acidic micro-droplets to produce sulfate and other oxidized sulfur species in the condensed phase, in the absence of any added oxidants other than $O_2$ (Hung and Hoffman, 2015). The observed process was fastest for droplets with a pH ~3, for which it was about 4 orders of





magnitude faster than S(IV) oxidation for typical ambient $H_2O_2$ levels. Several recent studies indicate a likely role for such heterogeneous $SO_2$ conversion to sulfate in major haze-fog events in polluted Chinese megacities (e.g. Xue et al., 2016, and references therein). For rapid sulfate production to affect the base:acid ratio in the present experiments, something must still have hindered the expected equilibration of the particles by uptake of DMA and ammonia. Since the $SO_2$ mixing ratio was

over three orders of magnitude larger than the DMA mixing ratio, the rates of nanoparticle collisions with $SO_2$ were more frequent than collisions with DMA by a similar factor. Therefore if 0.5% of $SO_2$-particle interactions resulted in the formation of non-volatile particle phase sulfur, this mechanism would outstrip the ability of DMA to neutralize the particles by over a factor of 5, resulting in particles that do not reach thermodynamic equilibrium with respect to the gas phase bases.

## 6 Conclusions

Nanoparticles were formed by reactions involving sulfuric acid, dimethylamine (DMA), water vapor, and contaminant ammonia. DMA is the more effective stabilizing base for sulfuric acid during the initial nucleation and growth to at least 2 nm, but at sizes around 10 nm and greater, ammonia was taken up into the particles with comparable or even greater efficiency. Given the greater abundance of ammonia than amines in most atmospheric regions, ammonia may be more important than DMA for forming particulate sulfate salts in growing nanoparticles. The sulfuric acid particles were not

observed to be fully neutralized by bases, despite the presence in the gas phase of at least an order of magnitude more DMA and $NH_3$ than $H_2SO_4$. For many particle collections at the smallest measurable particle sizes (around 10 nm), the base:acid ratio was a few times lower than for ~ 20 nm particles. This could be the result of a phase transition, in which particles moved from a supersaturated liquid to a phase-separated particle with a solid core as they grow. The base ions (ammonium and dimethylaminium) may have been maintained more stably in the solid phase of the larger particles upon sampling, and

we also consider it possible that the chemistry of particle growth was different for the particles of different phase states due to the different ion concentrations in their mobile phases. The very low particulate base:acid ratios observed do not have an unequivocal explanation, but they suggest that the sampled particles were not at thermodynamic equilibrium with the gas phase vapors. These observations suggest that effective stabilization of sulfuric acid in nanoparticles requires fewer base molecules than would be in a 2:1 or even a 1:1 salt. However, the very low base:acid ratios observed in these laboratory-

generated particles may not be expected under more typical ambient conditions of lower $SO_2$ and higher NOx.

## Acknowledgements

We would like to thank CERN for supporting CLOUD with important technical and financial resources, and for providing a particle beam from the CERN Proton Synchrotron. This research has received funding from the EC Seventh Framework Programme (Marie Curie Initial Training Network "CLOUD-ITN" grant no. 215072, the ERC-Advanced grant

"ATMNUCLE" (no. 227463), the German Federal Ministry of Education and Research (project no. 01LK0902A), the Swiss





National Science Foundation (project nos. 206621 125025 and 206620 130527), the Academy of Finland Centre of Excellence program (project no. 1118615), Academy of Finland (project no. 138951), the Austrian Science Fund (FWF; project nos. P19546 and L593), the Portuguese Foundation for Science and Technology (project no. CERN/FP/116387/2010), the US National Science Foundation, and the Russian Foundation for Basic Research (grant N08-02-91006- CERN). J.N.S. acknowledges funding from the Finnish Academy (Grant No. 251007) and U.S. Department of Energy (Grant No. DE-SC0014469). The National Center for Atmospheric Research is sponsored by the National Science Foundation.

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



| Run no. | DMA target | DMA | NH$_3$ | O$_3$ | SO$_2$ | UV aperture | H$_2$SO$_4$ |
| --- | --- | --- | --- | --- | --- | --- | --- |
| | ppt | ppt | ppt | ppb | ppb | % | x10$^7$ cm$^{-3}$ |
| 1032 | 40 | 51 | 22 | 23 | 63 | 100 | 1.3-1.4 |
| 1033 | 40 | (46) | (28) | 23 | 63 | 100 | 1.3 |
| 1035 | 40 | (46) | (28) | 23 | 63 | 40 | 0.5-0.8 |
| 1036 | 40 | (46) | (28) | 23 | 63 | 100 | 1.1-1.3 |
| 1040 | 10 | (23) | (19) | 23 | 63 | 20 | 0.4 |
| 1043 | 10 | (23) | (19) | 23 | 63 | 40 | 0.9-1.0 |
| 1047 | 10 | 28 | 28 | 86 | 63 | 100 | 2.4-2.7 |
| 1056 | 0 | <5 | <5 | 23 | 72 | 100 | 3.5 |

**Table 1. Overview of experimental conditions for the TDCIMS measurements described. The chamber temperature was 278 K and the relative humidity was 38%. DMA and NH$_3$ measurements are given (parts per trillion by volume) when coincident with the TDCIMS observations, and otherwise averages for the same experimental conditions close in time are given as estimates, indicated with parentheses. O$_3$ and SO$_2$ are reported in parts per billion by volume. Larger UV aperture fractions allow more ultraviolet light into the chamber, resulting in greater rates of SO$_2$ oxidation to H$_2$SO$_4$.**



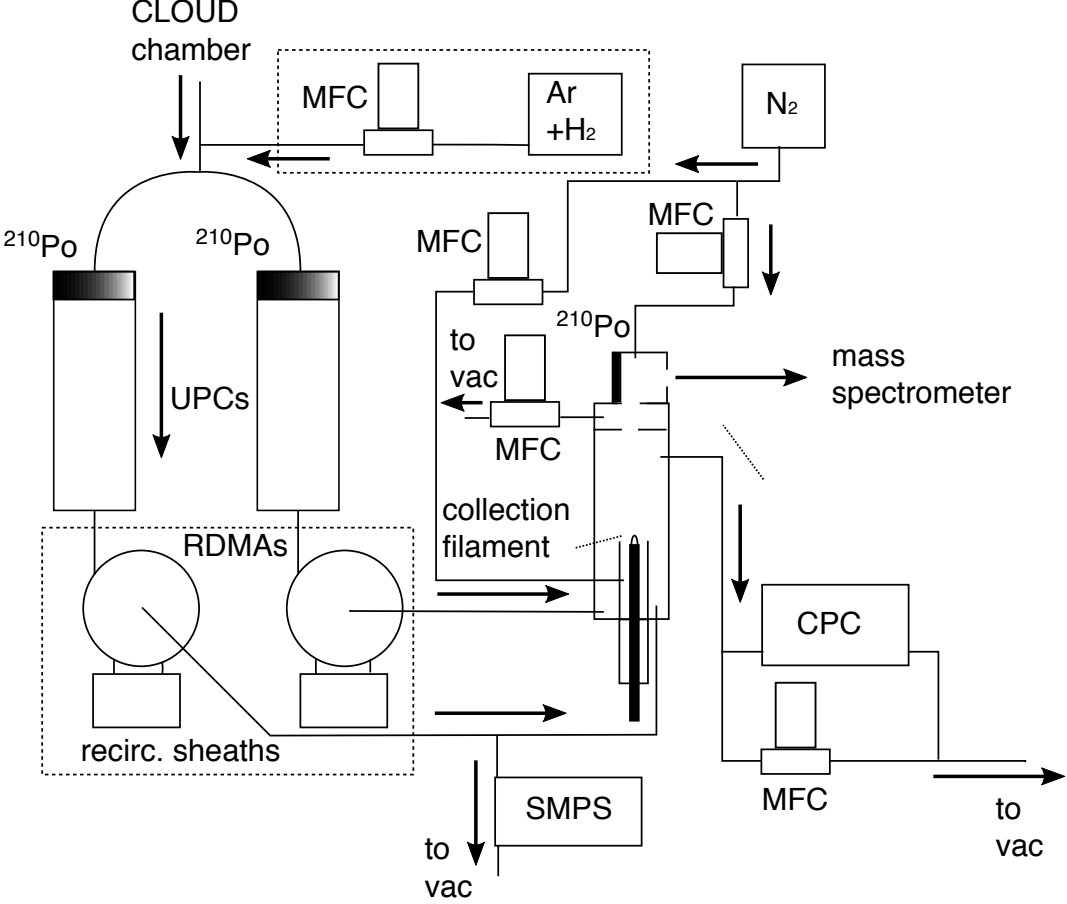

**Figure 1. Schematic of the TDCIMS instrument as deployed at CLOUD7. Dashed boxes show components that were not always connected. Particles from sampled chamber air were negatively charged in the UPCs (unipolar chargers), optionally size-selected**
5 **by radial differential mobility analyzers (RDMAs), and electrostatically precipitated onto a Pt collection filament (shown here in collection position) held at +4kV over a period of 15 or 30 minutes. The filament was then translated vertically into the ion source of a time-of-flight (TOF) mass spectrometer and the sampled material was thermally desorbed and/or decomposed by a temperature ramp of the wire at atmospheric pressure in a dry $N_2$ environment. The resulting compounds were ionized and passed into the TOF via electrostatic and octopole ion guides. CPC: condensation particle counter. MFC: mass flow controller. Vacuum**
10 **pumps and vacuum lines have been omitted from the drawing.**





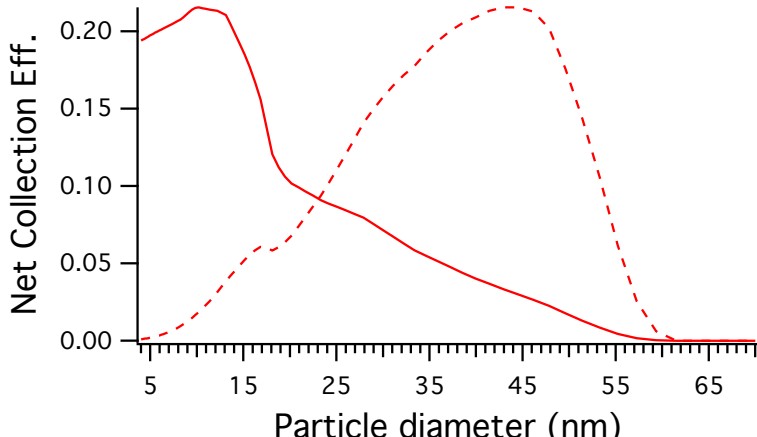

**Figure 2. Solid line: Estimated TDCIMS particle sampling efficiency as a function of mobility diameter for bulk (non-size-resolved) sampling. Dashed line: the same number efficiency scaled by the volume of an individual particle at each size and normalized to fit on the same plot (arbitrary units). These estimates include the effects of sampling losses, (multiple) charging efficiency, and efficiency of capture by the electrostatic precipitator. Small (<15 nm) particles are efficiently captured due to their high mobility. The efficiency of collection at sizes below 4 nm could not be assessed using the available experimental apparatus, but particles at smaller sizes are not likely to have contributed detectable amounts of particle mass during these experiments.**





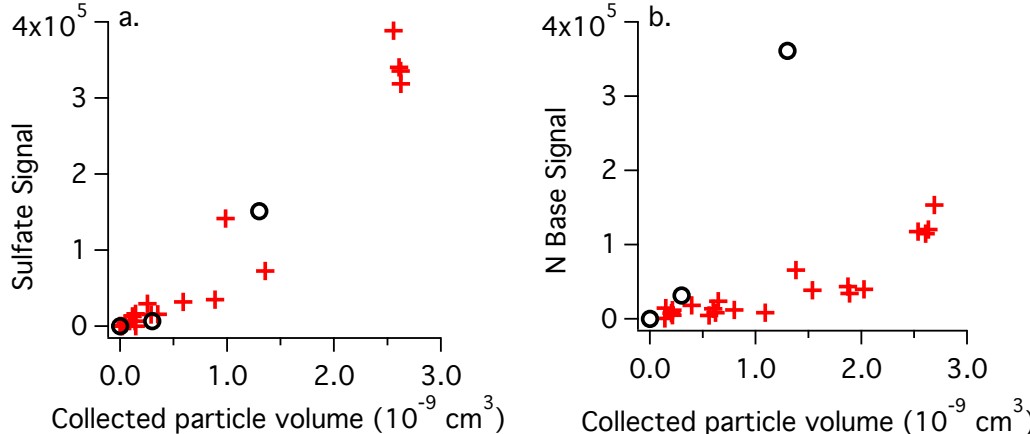

**Figure 3. a.** Sum of oxidized sulfur ("sulfate") signal vs. collected particle volume. **b.** Sum of N base (DMAH$^+$ and NH$_4^+$) signal vs. collected particle volume. Red crosses from chamber particle collections of DMA + H$_2$SO$_4$ nucleation experiments, and black circles are from calibration ammonium sulfate aerosol. Ion counting errors are smaller than the symbols and are excluded. The negative ion data from the chamber particles are broadly consistent with particles dominated in volume by sulfate. However, the chamber particles have a much lower base fraction than the ammonium sulfate calibration aerosol.





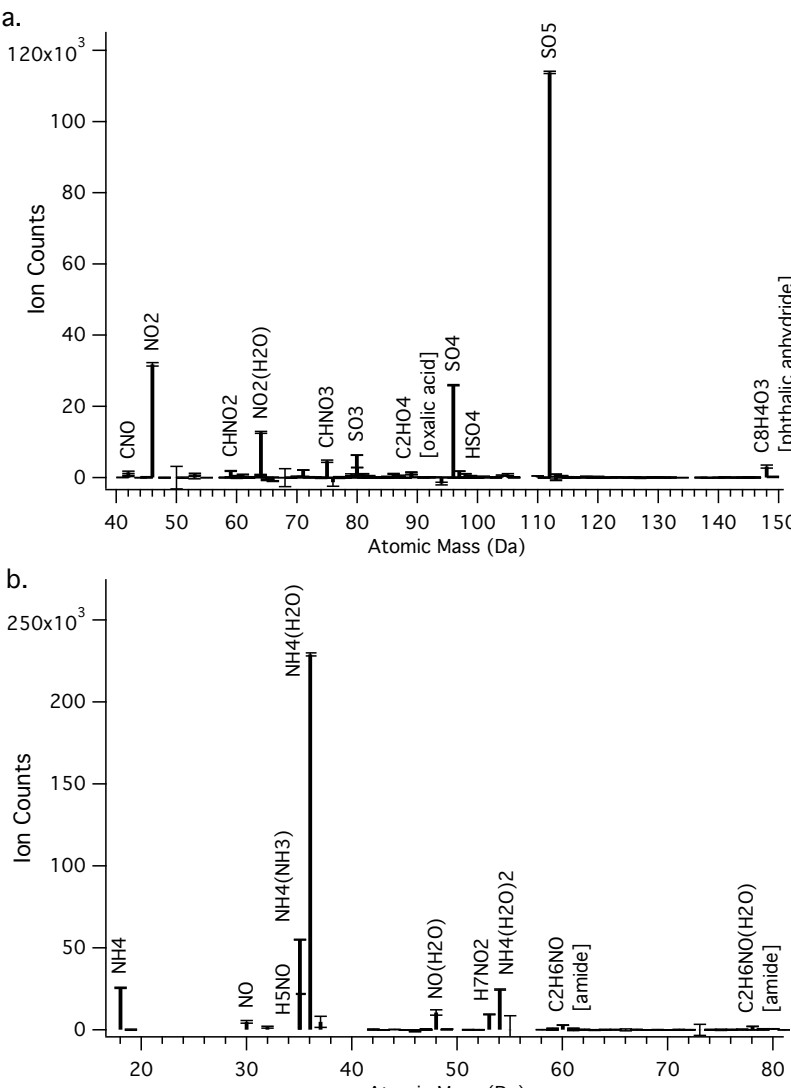

**Figure 4.** Calibration ammonium sulfate aerosol mass spectrum for (a.) negative and (b.) positive ions. $SO_5^-$ and $SO_4^-$ are the main sulfate ions, and $NO_2^-$ indicates a nitrate contaminant in the standard, most likely introduced by the water or air used for the atomization. $NH_4^+$ and its clusters with water and $NH_3$ are the main ammonium ions. All species plotted have one elemental charge.





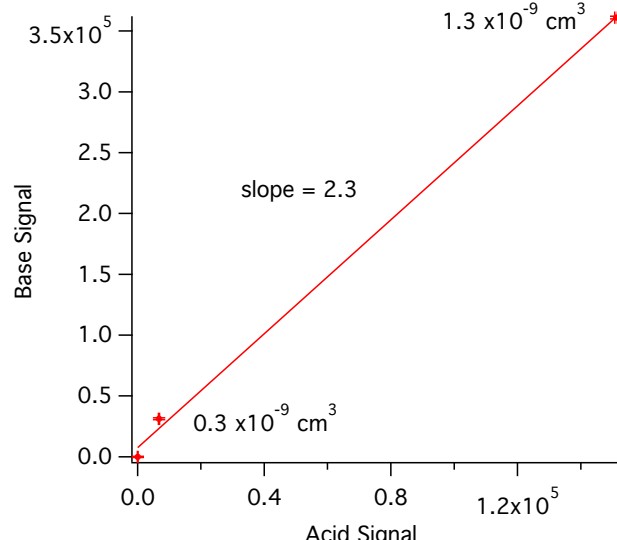

**Figure 5. TDCIMS calibration results for atomized ammonium sulfate solution during the CLOUD7 campaign. Estimated collected particle volumes are given. For ammonium sulfate aerosol during CLOUD7, the TDCIMS was more sensitive to ammonium than to sulfate on a molar basis (2.3:1 ion signal for the 2:1 maximum expected ammonium to sulfate ratio in the salt). One standard error bars of ion signal are shown but they are small with respect to the symbols.**



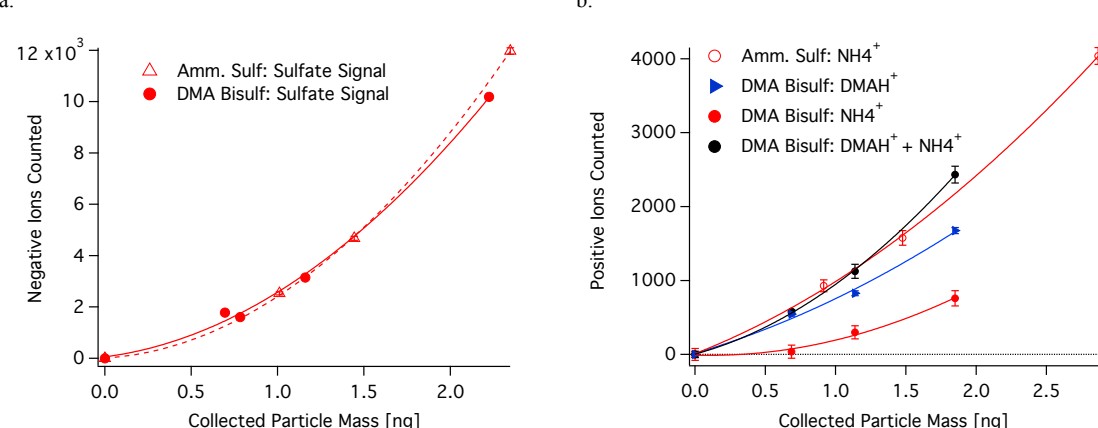

**Figure 6. Results of a post-CLOUD7 calibration of the TDCIMS with salts generated by atomizing liquid solutions of ammonium sulfate, (NH₄)₂SO₄ and dimethylaminium bisulfate (DMABS), (C₂H₈N)HSO₄. Particles of roughly 25 nm volume median diameter**

5 **were used for this calibration, and larger mass collections were achieved by collecting for longer periods. For these calibrations, particle volume was corrected to mass using the densities of DMAS (1.35 g cm⁻³) and AS (1.78 g cm⁻³). (a.) Signal in negative ion mode plotted against particle mass, with a second-order polynomial fit. The sulfate sensitivity for the two salts appears to be indistinguishable. (b.) Signal in positive ion mode for the two salts. There was contamination of the DMABS by ammonium at some point between solution generation and analysis. Based on laboratory tests, this most likely occurs as a result of trace**

10 **ammonia present in the carrier N₂ and transport lines. For this reason, the DMAH⁺ signal, NH₄⁺ signal, and their sum are all given. The sum signal for DMABS is very similar to the AS signal, suggesting that both aerosols may actually have similar base:acid ratios after atomization and drying.**





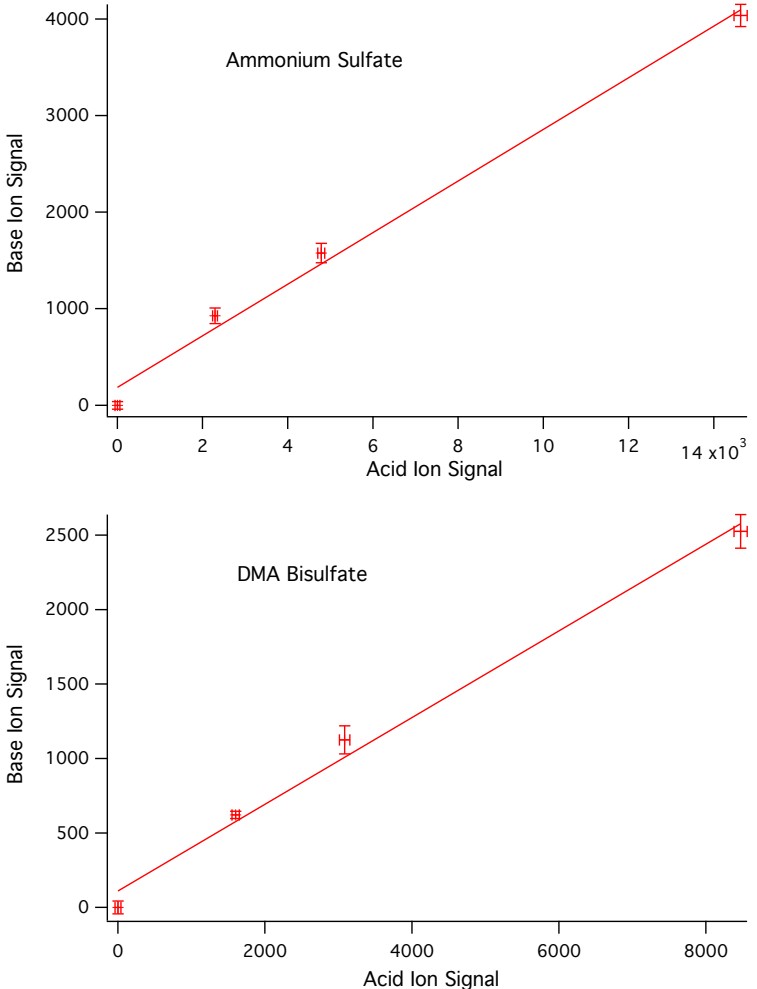

**Figure 7.** Base (DMAH$^+$ + NH$_4^+$) ion signals plotted against sulfate ion signals for ammonium sulfate and dimethylaminium bisulfate calibration aerosol over estimated sample loadings of 0 to 1.8 ng, conducted in the laboratory at NCAR after CLOUD7
5  (the same tests as presented in Figure 6). Signals were scaled by collected mass to match positive and negative analyses. Despite the nonlinear character in the individual ions' dependence on the sample mass (Figure 6), the base:acid ratio remains constant over a range of collected masses for the same calibration aerosol, in agreement with the ammonium sulfate aerosol calibration performed at CLOUD7 (Figure 5). Note that relative sensitivities to base and acid were different during this test than during the calibrations at CLOUD7 (Figure 5), which should be considered the better comparison for the experimental particle data since they were
10  conducted under CLOUD7 conditions.




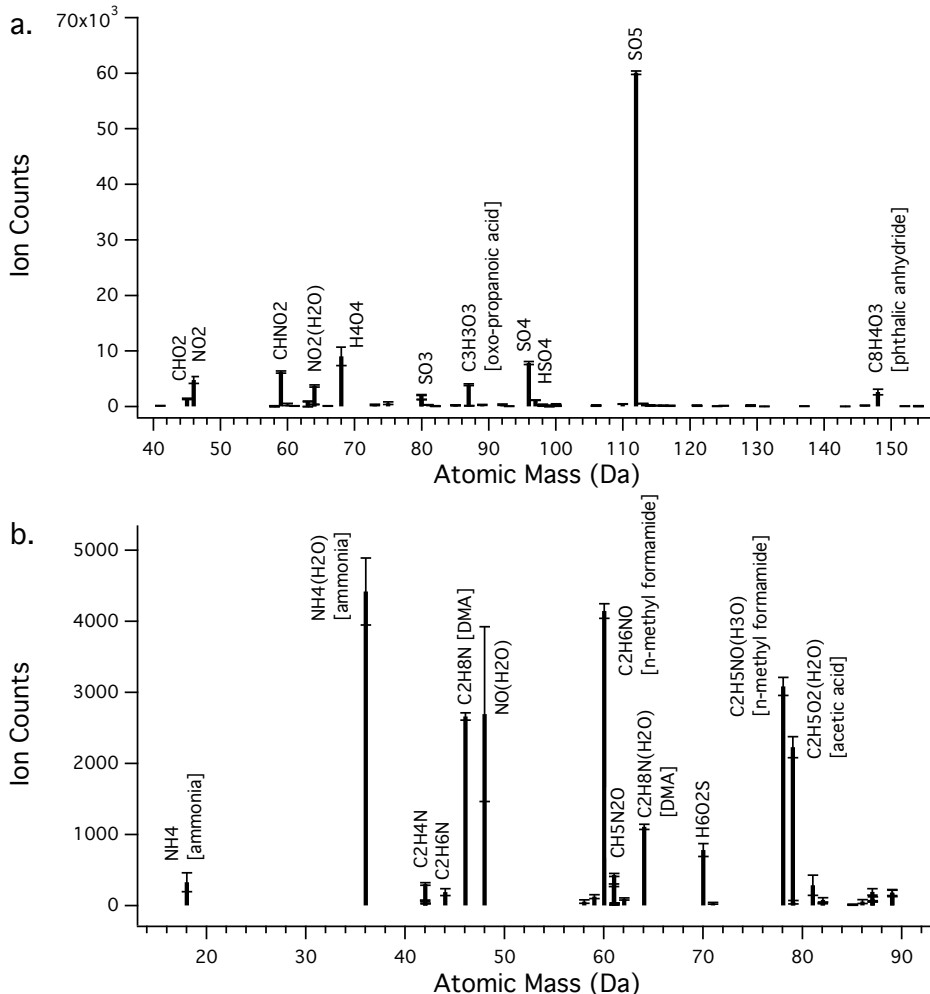

**Figure 8.** Negative (a.) and positive (b.) background-corrected 30-minute collection particle mass spectra for a particle formation experiment with nominal 40 ppt DMA, 23.5 ppb $O_3$, and $2.8 \times 10^6$ cm$^{-3}$ cm$^{-3}$ $H_2SO_4$, run 1033 on October 16-17, 2012. All species plotted have one elemental charge. The volume median diameter of collected particles was 10.9 nm for positive ions and 13.1 nm for the negative ions. Particulate ammonium and dimethylaminium levels were similar, despite not adding gas phase ammonia intentionally to the chamber. N-methyl formamide was a contaminant generated in the particle chargers. The negative spectrum is dominated by the sulfate-derived peak $SO_5^-$.



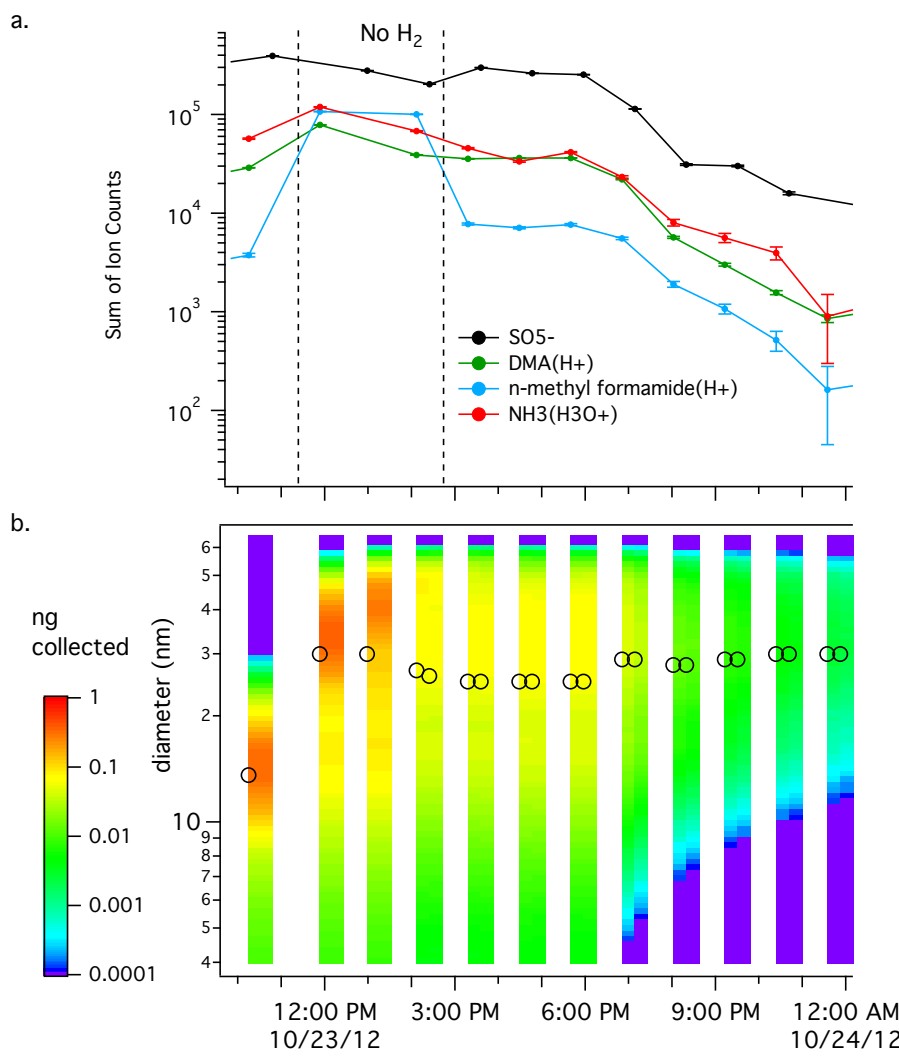

**Figure 9. a. Particulate ion signals for sulfate (SO$_5^-$), dimethylamine, n-methyl formamide, and ammonia during a nucleation experiment with 10 ppt DMA and 2.6x10$^7$ cm$^{-3}$ H$_2$SO$_4$ (run 1047). For most of this run, 1% H$_2$ was added to the TDCIMS inlet as a gas phase radical scavenger, except for the period between the dashed vertical lines. During this period, the n-methyl formamide increased about 30-fold, while the other species underwent more modest changes. b. Size-resolved particle mass collected during this period. White gaps indicate background periods when no particles were collected. Open circles show the volume mean diameter for collected particles.**



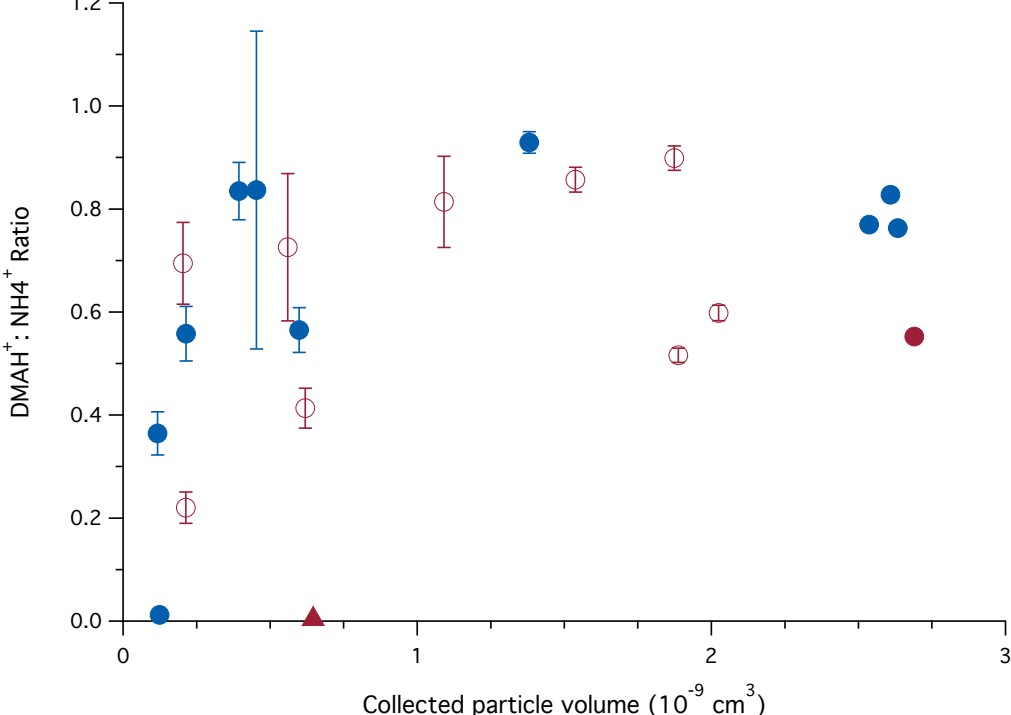

**Figure 10.** The ratio of summed DMAH$^+$ to summed NH$_4^+$ ion signals plotted against the collected particle volume for DMA+H$_2$SO$_4$ particle formation experiments (circles) and for the H$_2$SO$_4$ only (no added DMA) nucleation experiment (triangle). Collections of less than 0.1 x 10$^{-9}$ cm$^{-3}$ of sample were excluded due to low signal-to-noise. The filled symbols are periods with nominally 10 ppt DMA in the chamber and the open symbols are with 40 ppt DMA. Blue symbols indicate periods with 1% H$_2$ added to the TDCIMS inlet to scavenge radicals, and red symbols are periods without H$_2$. The vertical error bars indicate the standard error of the ion signals. Higher ammonium fractions were more likely for low collected masses (resulting from low chamber aerosol mass), possibly because less contaminant ammonia was required to alter the composition of the particles. DMAH$^+$ was never observed to be more abundant than NH$_4^+$ in the particles during any of experiments.





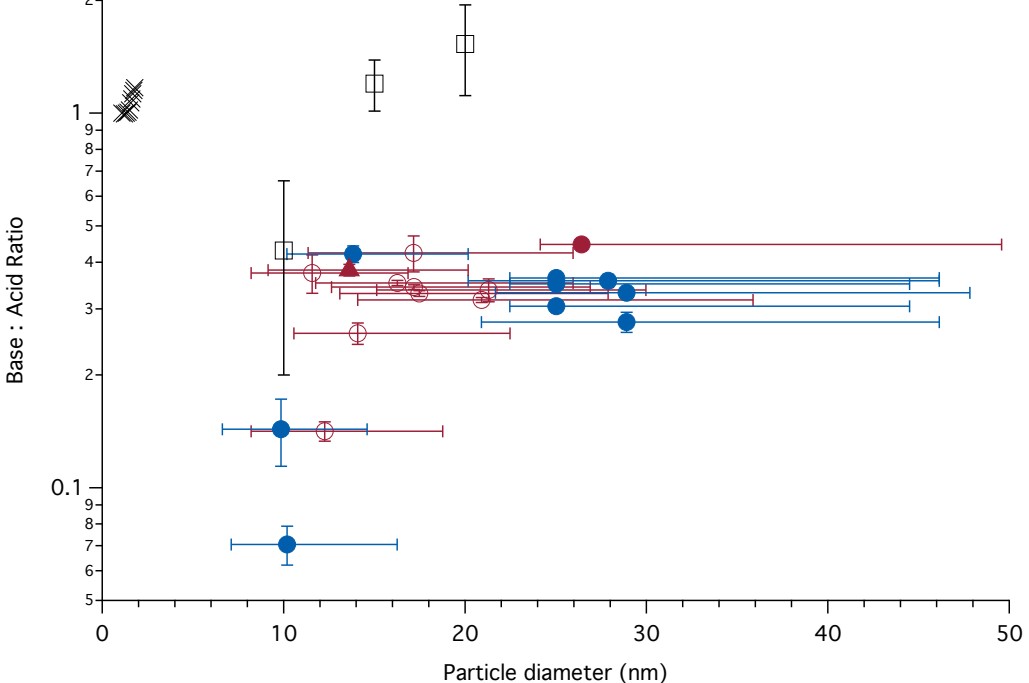

**Figure 11. Base:acid TDCIMS signal ratios for DMA-H$_2$SO$_4$ experiments (circles) and the H$_2$SO$_4$ alone experiment (triangle) plotted against the volume median particle diameter for collected particles; signal-weighted APiTOF positive ion cluster composition (black X markers) (Bianchi et al. 2014); HTDMA-based estimates (black squares) assuming 1:1 ammonium:dimethylaminium ratio, with standard deviations (n= 3-6) from Ahlm et al. (2016) and Kim et al. (2016). The TDCIMS base:acid ratio is the summed NH$_4^+$ and DMAH$^+$ ions and their clusters divided by summed sulfate peaks. Collections of less than 0.1 x10$^{-9}$ cm$^3$ of material were excluded. Only experiments for which the collected mass for the two polarities were within a factor of 2 are included in the plot, and the ratios are scaled to correct for differences in sampled mass. The filled symbols are experiments with nominally 10 ppt DMA and the open symbols are with 40 ppt DMA. Blue symbols indicate periods with 1% H$_2$ added to the TDCIMS inlet and red symbols are without H$_2$. The vertical error bars represent the standard error of the particle ion signals, and the horizontal error bars indicate the range of particle diameters that contributed to the collected mass (minimum 2% contribution). The TDCIMS results show consistently lower base:acid ratios than inferred by the HTDMA, but they show a similar size dependence and high variability for the smallest detectable sizes.**