# Peer review of "Unexpectedly acidic nanoparticles formed in dimethylamineammonia-sulfuric acid nucleation experiments at CLOUD"

_Atmospheric Chemistry and Physics, 2016_

## Referee Comment (RC1) · Anonymous Referee #1 · 26 Jul 2016

This paper explored the chemical composition of newly formed nanoparticles (2-30 nm) from dimethylamine-ammonia-sulfuric acid using CLOUD chamber experiment. This study used the Thermal Desorption Chemical Ionization Mass Spectrometer (TDCIMS) to measure the composition of newly formed nanoparticles ($\sim$10 nm). The resulting data of this paper reported that the base to acid ratio in the small particles (less than 10 nm) was lower than thermodynamically predicted values and never reach to entire neutralization of sulfuric acid even with the excess amount of base gases. When particles reached to $\sim$20 nm particles, the base to acid ratio accorded with thermodynamic compositions. This paper concluded that the particles less than 10 nm were more acidic than expected as shown in measured aerosol compositions. The authors

suggested that in the small particle size range (∼10 nm), the rapid heterogeneous conversion of SO2 to sulfate does not need to reach thermodynamic equilibrium with respect to the bases, while it happens to be in the bulk phase (or larger size particles). Overall, no clear conclusion appeared for the new acidic particle formation in the presence of base gas due to the lack of data points and contamination. Although the observation of this paper may not be expected under the ambient conditions of low SO2 and high NOx, the authors also need to provide meaningful implication of the resulting data to the ambient environment. This paper requires major revision. Please find the comments below.

1. Overall, pages 1-10 are related to experimental methods and instrumentations. The actual discussion using laboratory data appeared between pages 11-16. I would like to ask authors to reduce experimental and instrumental sections and focus more on the application of the results to the ambient atmosphere.

2. Introduction section. In general, the amine concentration is much lower than ammonia in ambient air. The authors should provide rationale or hypothesis for why new particle formation experiment from SO2 oxidation has been conducted in the presence of amine. It is difficult to clearly understand the major goal of this paper in the introduction.

3. The effect of humidity on new particle formation should be explained in depth. In page 3, the authors showed that relative humidity (RH) was constant at 38%. Although RH is fixed in this paper, RH is an important parameter to determine the phase of inorganic salt aerosol and liquid water content. Subsequently, the aerosol phase and water content will also influence new particle formation from SO2 oxidation. For example, efflorescent RH (ERH) of ammonium sulfate is near 40%. Is there a specific reason to select RH at 38% ?

4. In Table 1. Ozone was included in the chamber. Ozone is known to influences heterogeneously SO2 oxidation and increase sulfuric acid formation. What is the potential effect of ozone on the new particle formation? Is there a specific reason to fix at 23ppb? Run1047 was operated at the higher ozone concentration but no clear explanation appeared.

5. Figure 4. There were some other peaks from oxalic acid and nitrate (nitrite ?). How do the authors know that the neutralization of nitrate with ammonia can reduce the base-to-acid ratio (amine + ammonia vs. sulfate) ? Page 15, line 31, the author pointed out that the NOx concentration of CLOUD chamber was low. For the Figure 4 and 8, the NOx contamination was not negligible. The NOx effect may source large uncertainty for the final results.

6. Page 6, line 31 and Figure 5. There are limited number of data in the middle range $(0.4 \times 105 \sim 1.2 \times 105)$ to correlate linearly between acid and base signals.

7. Page 9, line 13-14. What is the source of oxalic acid?

8. Figure 10, it would better to explain ammonium contribution to aerosol compositions in the text. Sentence "...possibly because less contaminant ammonium was required..." in Figure caption should be informed to readers to better understand the results.

9. Page 11, line 10-12. The authors need to discuss more clearly why ammonia becomes abundant at the particle larger than 10 nm. What is the physical state of inorganic salt aerosol (NH4-amine-sulfuric acid) at a given RH? If the inorganic salt aerosol is solid, the heterogeneous reaction of SO2 on the salt aerosol will be less efficient that in the aerosol below the ERH and furthermore uptake of amine and ammonia would be affected by physical state. What is the influence of RH on the chemical compositions of aerosol?

10. Page 12, line 9. Please explain why nanoparticle is liquid. The study by Cheng et al. (2015) suggested that nanoparticles tend to be aqueous phase at room temperature. The temperature of For the CLOUD chamber experiment, was much lower than

room temperature. What is the impact of temperature on aerosol phase ?

11. Page 14, line 13 and Figure 3. The data from Figure 3 are not sufficient to prove that sulfate signals are consistent with sampled particle mass.

12. Page 16, line 8, the authors mentioned about the artifact of DMA on SO2-particle interaction and yielded the factor of 5 for DMA system. The authors need to provide more detailed explanation for how to get the factor of 5.

13. Page 5, line 32, "to obtain agreement of the chamber" should be "to obtain the agreement of the chamber".

14. Page 6, line 1, "determine the sizes and total volume" should be "determine the size and the total volume".

15. Page 6, line 5, "there was" should be "there were".

16. Page 12, line 14, "a more stable salt with sulfuric acid than is ammonia", delete "is".

17. Page 13, line 25, "clusters may is acidic for" should be "clusters may be acidic for"

18. Page 14, line 20, "(Kim et al., 2016)" this citation should belong to next sentence.

---

## Referee Comment (RC2) · Anonymous Referee #2 · 31 Jul 2016

This is a nicely performed study with very clearly explained measurements. My only suggestions are for revision of the discussion section, where I think the authors may be speculating a bit too much on the implications of their data:

1. Section beginning with: "The TDCIMS data overall indicate a large increase of base fraction as the particles grew from 10 to 20 nm." I think the authors should be bit more cautious in interpreting their data in this section. Aside from 3 data points, it appears that the base:acid ratios in figure 11 are bound between 0.2 and 0.5 for all measurements. While the results presented are very interesting I think it is premature to conclude that the chemical composition of particles is unquestionably changing as they grow from 10-20 nm.

2. Similarly, there is no evidence in the measurements reported here for a phase transition in particles as they grow from 10-20 nm. They do show that 10-20 nm particles sampled are more acidic than basic, and are composed of slightly more ammonia than dimethylamine. More analysis seems necessary to examine the phase of the detected particles.

Minor Comment:

1. The manuscript refers to "Kim et al 2016". However, this citation does not appear in the references section. Also, I do not believe the "Alhm et al, 2016" citation is published in Aerosol Science and Technology (perhaps it is in review).

---

## Author Comment (AC1) · 7 Sep 2016

Response to Anonymous Referee # 1

This paper explored the chemical composition of newly formed nanoparticles (2-30 nm) from dimethylamine-ammonia-sulfuric acid using CLOUD chamber experiment. This study used the Thermal Desorption Chemical Ionization Mass Spectrometer (TDCIMS) to measure the composition of newly formed nanoparticles (~10 nm). The resulting data of this paper reported that the base to acid ratio in the small particles (less than 10 nm) was lower than thermodynamically predicted values and never reach to entire neutralization of sulfuric acid even with the excess amount of base gases. When particles reached to ~20 nm particles, the base to acid ratio accorded with thermodynamic compositions. This paper concluded that the particles less than 10 nm were more acidic than expected as shown in measured aerosol compositions. The authors suggested that in the small particle size range (~10 nm), the rapid heterogeneous conversion of SO2 to sulfate does not need to reach thermodynamic equilibrium with respect to the bases, while it happens to be in the bulk phase (or larger size particles). Overall, no clear conclusion appeared for the new acidic particle formation in the presence of base gas due to the lack of data points and contamination. Although the observation of this paper may not be expected under the ambient conditions of low SO2 and high NOx, the authors also need to provide meaningful implication of the resulting data to the ambient environment. This paper requires major revision. Please find the comments below.

*This review of our findings is mostly accurate, but we point out that even for 20 nm particles, the TDCIMS observations show more acidic particles than thermodynamically expected. The HTDMA results are consistent with thermodynamic expectations for ~20 nm particles, so there is some discrepancy between the two instruments, as discussed in the manuscript. We appreciate the reviewer's comment and have added to the discussion of implications for the ambient environment (see response to comment 1 below).*

1. Overall, pages 1-10 are related to experimental methods and instrumentations. The actual discussion using laboratory data appeared between pages 11-16. I would like to ask authors to reduce experimental and instrumental sections and focus more on the application of the results to the ambient atmosphere.

*We aimed to be as thorough as possible with the methodology section because of the unexpected nature of the results and the fact that this is the first application of this technique to this chemical system. Referee #2 appeared to appreciate this balance. We will remove Figure 5 on the basis of this comment and comment 6.*

*We added this section discussing the application to the ambient atmosphere:*

*"5.3 Implications for ambient nanoparticles*

*"Newly formed nanoparticles in ambient air in the planetary boundary layer are unlikely to contain only sulfuric acid and nitrogen bases once they reach the sizes considered in this study. However, by observing this chemically simple system, it was possible to probe what appears to be a rapid heterogeneous reaction which may play a role in some atmospheric regions. If the inferred reaction probability for $SO_2$ is applicable in other circumstances, then this reaction is competitive with $H_2SO_4$ uptake when the $SO_2$ concentration is about a factor of 2000 times larger*

*than the H$_2$SO$_4$ concentration (e.g. about 1 ppb SO$_2$ and 1x10$^7$ cm$^{-3}$ at sea level). This is most likely to occur at night in regions with strong SO$_2$ emissions, when short-lived, photochemically produced H$_2$SO$_4$ is at low concentrations but SO$_2$ levels remain high. This is consistent with the findings of Xue et al. (2016), who show that a diurnally unvarying SO$_2$-surface reaction is likely an important contributor to secondary aerosol formation in haze-fog events in Chinese megacities, and that it represents a significant fraction (about a third) of the sulfate production at night. SO$_2$ levels for these conditions were similar to the present study, in the range of tens of ppb. For comparison, if we assume a 50 ppb SO$_2$ mixing ratio and a particle distribution for which the available particle phase surface area is represented by 100 nm diameter condensation nuclei with a concentration of 5000 cm$^{-3}$, a 0.05% SO$_2$-surface reaction probability leads to a 5 µg m$^{-3}$ sulfate production rate, which is of the right order of magnitude to explain the polluted megacity observations. However, this surface mechanism is much less likely to be important for atmospheric new particle formation in remote, unpolluted areas. For example, at the well-studied Hyytiälä site in the Finnish boreal forest, SO$_2$ concentrations around 0.1 ppb are typical. At this low level, the SO$_2$-surface reaction could only contribute about 0.1 nm hr$^{-1}$ to growth, which is a small fraction of typical growth rates observed at this site (Yli-Juutti et al., 2011)."*

2. Introduction section. In general, the amine concentration is much lower than ammonia in ambient air. The authors should provide rationale or hypothesis for why new particle formation experiment from SO2 oxidation has been conducted in the presence of amine. It is difficult to clearly understand the major goal of this paper in the introduction.

*We added this text to the introduction:*

*"While ammonia is present at much higher concentrations than DMA in the remote atmosphere, amines such as DMA can effectively compete with ammonia in the growth of nanoparticles of ~10 nm diameter (Smith et al., 2010; Barsanti et al., 2009)."*

*Smith, J. N., Barsanti, K. C., Friedli, H. R., Ehn, M., Kulmala, M., Collins, D. R., Scheckman, J. H., Williams, B. J., and McMurry, P. H. Observations of aminium salts in atmospheric nanoparticles and possible climatic implications. Proceedings of the National Academy of Sciences of the United States of America, 107(15), 6634–9, doi:10.1073/pnas.0912127107, 2010.*

*Barsanti, K., McMurry, P. H. and Smith, J. N.: The potential contribution of organic salts to new particle growth, Atmos. Chem. Phys., 9, 2949–2957, 2009.*

3. The effect of humidity on new particle formation should be explained in depth. In page 3, the authors showed that relative humidity (RH) was constant at 38%. Although RH is fixed in this paper, RH is an important parameter to determine the phase of inorganic salt aerosol and liquid water content. Subsequently, the aerosol phase and water content will also influence new particle formation from SO2 oxidation. For example, efflorescent RH (ERH) of ammonium sulfate is near 40%. Is there a specific reason to select RH at 38%?

*The presence of water vapor is clearly important to the particle formation. We also added this statement and reference to the Introduction:*

*"…and water vapor has been shown to be an important contributor to the initial steps of new*

*particle formation (e.g. Zollner et al., 2012, and references therein).*

*Zollner, J. H., W. A. Glasoe, B. Panta, K. K. Carlson, P. H. McMurry, and D. R. Hanson. 2012. "Sulfuric Acid Nucleation: Power Dependencies, Variation with Relative Humidity, and Effect of Bases." Atmospheric Chemistry and Physics 12 (10): 4399–4411. doi:10.5194/acp-12-4399-2012.*

*The phase of the particles is certainly of key importance in this discussion. We discussed this already in the submitted manuscript, but in response to this and other comments from Referee 1, we added details to improve understanding about this and its role in SO2 uptake and oxidation. In particular, see response to comment 9, and also to comments 4 and 10.*

*There is not a reason why the RH was 38% specifically, other than for consistency with previous studies in the CLOUD chamber (e.g. Kirkby et al. 2011) and that it is relevant to the lower atmosphere.*

4. In Table 1. Ozone was included in the chamber. Ozone is known to influences heterogeneously SO2 oxidation and increase sulfuric acid formation. What is the potential effect of ozone on the new particle formation? Is there a specific reason to fix at 23ppb? Run1047 was operated at the higher ozone concentration but no clear explanation appeared.

*Ozone was set to 23 ppb as a value relevant to the troposphere, and sufficient for generating OH to form H2SO4 at necessary levels. We added these sentences to the Introduction:*

*"The chamber conditions were set to simulate atmospheric conditions relevant to the lower troposphere."*

*"The ozone mixing ratio was increased to 86 ppb for one experiment to increase $H_2SO_4$ levels and the new particle formation rate."*

*In the original text, a discussion of aqueous phase oxidation by O3 and H2O2 was included (page 15, lines 23-31). In the updated manuscript we include more detailed information on possible aqueous phase oxidation in the experiments:*

*"For particles of pH < 4 and for low liquid water content, as in the case of the recently formed particles in this study, $H_2O_2$ is the dominant oxidant of S(IV) in the atmosphere. To estimate the potential of aqueous phase S(IV) oxidation to contribute to nanoparticle growth, we considered the rate of $H_2O_2$ + S(IV) reaction in a 5.4 nm diameter ammonium bisulfate particle as a proxy for the chamber particles. We used the Extended Aerosol Thermodynamics Model (E-AIM) to estimate the pH and assumed that [$SO_2$] and [$H_2O_2$] were in Henry's Law equilibrium with the gas phase. For a gas phase $H_2O_2$ level of 1 ppb, S(IV) oxidation was estimated at about 0.002 molecules $s^{-1}$ for the small particle. For comparison, $H_2SO_4$ collisions with the particle would occur at a rate of about 0.2 molecules $s^{-1}$ for an $H_2SO_4$ concentration of $1x10^7$ $cm^{-3}$, with still larger contributions from cluster collisions and coagulation. This aqueous chemistry was therefore an insignificant contributor to the particle growth and composition."*

5. Figure 4. There were some other peaks from oxalic acid and nitrate (nitrite ?). How do the authors know that the neutralization of nitrate with ammonia can reduce the base-to-acid ratio

(amine + ammonia vs. sulfate)? Page 15, line 31, the author pointed out that the NOx concentration of CLOUD chamber was low. For the Figure 4 and 8, the NOx contamination was not negligible. The NOx effect may source large uncertainty for the final results.

*The instrument is more sensitive to ammonium nitrate than to ammonium sulfate by a factor of about 100, so we added this to the captions:*

*"The instrument is more sensitive to ammonium nitrate (detected as $NO_2^-$) than to ammonium sulfate by a factor of ~100 (Bzdek et al., 2014), so the nitrate contamination is likely a small fraction of the sampled aerosol mass."*

6. Page 6, line 31 and Figure 5. There are limited number of data in the middle range (0.4×105 ~ 1.2×105) to correlate linearly between acid and base signals.

*We know from tests in lab (Figure 7) that the base:acid ratio response is linear over the relevant range of sampled mass. The point of Figure 5 is mainly to demonstrate the relative sensitivities of acid and base during the campaign. We will remove the figure and state this in the text:*

*"These AS aerosol calibrations indicate that the TDCIMS was roughly equally sensitive on a molar basis to ammonium and sulfate during CLOUD7, with a 2.3:1 ammonium to sulfate signal ratio for AS.*

7. Page 9, line 13-14. What is the source of oxalic acid?

*Oxalic acid was only indicated in Figure 4, not in the text, though other organic contaminants were mentioned in the text. Oxalic acid was a minor contaminant in the calibration aerosol. In the text, oxo-propanoic acid was mentioned, and this was apparently a contaminant present in the chamber. We specified that they are contaminants in the text and do not have further information about their sources, so we do not see a need to change the manuscript.*

8. Figure 10, it would better to explain ammonium contribution to aerosol compositions in the text. Sentence "...possibly because less contaminant ammonium was required. . ." in Figure caption should be informed to readers to better understand the results.

*We interpret this comment to mean that the reader should get more of the interpretation from the main text, rather than the figure caption. We added "(see Results)" to the end of this sentence.*

9. Page 11, line 10-12. The authors need to discuss more clearly why ammonia becomes abundant at the particle larger than 10 nm. What is the physical state of inorganic salt aerosol (NH4-amine-sulfuric acid) at a given RH? If the inorganic salt aerosol is solid, the heterogeneous reaction of SO2 on the salt aerosol will be less efficient that in the aerosol below the ERH and furthermore uptake of amine and ammonia would be affected by physical state. What is the influence of RH on the chemical compositions of aerosol?

*The distinction here with respect to ammonia is really between the "molecular clusters" and "observable-by-TDCIMS nanoparticles", not between different sizes which are observable by TDCIMS and HTDMA, so we removed the ">10 nm" qualifier to make this clearer.*

*We agree that the phase state is critical, and we discuss this later in the paper (especially page 15, lines 6-21), where we bring up the fact that the observed particles in this size range may span the transition from liquid to mixed-phase (solid + liquid). Knowing the phase state is more difficult, and even for the extremely well-studied ammonium sulfate salt, it took a recent detailed theoretical-observational study to characterize its size-dependent phase state (Cheng et al., 2015). Such an analysis is not yet available for a mixed ammonium-aminium-sulfate system. We added statements to better constrain what we think we can infer about the phase state:*

*"…these small particles should have a liquid phase (Cheng et al., 2015). Supersaturated ammonium sulfate nanoparticles, such as those likely to be produced in new particle formation, are expected to be liquid or mixed-phase (depending on particle size) at the temperature and sizes considered in these experiments (see Discussion for more details)."*

*"At the slightly lower temperature in the present study (273 K), this [phase] transition [for ammonium sulfate] occurs at a slightly smaller particle diameter, closer to 9 nm. For the dimethylammonium-ammonium-sulfate nanoparticles studied here, a similar size-dependent transition seems plausible. However, less is known about the thermodynamics of these mixed ammonium-aminium sulfate salts. Even for large droplets, where size-dependent effects are not important for phase state, pure dimethylaminium sulfate does not form a solid phase, even at < 3 % RH. However, the exposure of such an amorphous liquid droplet to ammonia results in the formation of a solid phase which contains ammonium and aminium and is resistant to further exchange (Chan and Chan, 2013)."*

*We added this statement to the Conclusion in recognition of the need for further work in this area:*

*"However, further work is needed to characterize the phase states of mixed ammonium-aminium-sulfate nanoparticles to assess these possibilities and to understand the growth pathways of such particles."*

10. Page 12, line 9. Please explain why nanoparticle is liquid. The study by Cheng et al. (2015) suggested that nanoparticles tend to be aqueous phase at room temperature. The temperature of For the CLOUD chamber experiment, was much lower than room temperature. What is the impact of temperature on aerosol phase?

*Cheng et al. showed that even at very low temperatures (e.g., 215 K in their figure), ammonium sulfate (AS) particles are liquid or mixed-phase at sizes below about 20 nm. The experiments in this study were conducted at 273 K, for which temperature the AS nanoparticle phase transition occurs around 9 nm. We now state that we expect the particles to have a liquid phase under these size and temperature conditions, not that they are necessarily fully liquid. Please see response to comment 9 for further information and the relevant text which was added to the manuscript.*

11. Page 14, line 13 and Figure 3. The data from Figure 3 are not sufficient to prove that sulfate signals are consistent with sampled particle mass.

*We modified the statement:*

*"…there was no evidence for increased sulfate signal in chamber samples with respect to calibration standards of similar sampled mass."*

12. Page 16, line 8, the authors mentioned about the artifact of DMA on SO2-particle interaction and yielded the factor of 5 for DMA system. The authors need to provide more detailed explanation for how to get the factor of 5.

*We are not certain what the referee means by the "artifact of DMA on SO2-particle interaction". However, we modified the text to clarify the nature of the calculation performed:*

*If 0.05% of $SO_2$ collisions with a 5 nm particle irreversibly formed particle phase sulfate, even the largest discrepancies between predicted and observed growth rates could be reconciled. This hypothetical rate would be about three times as fast as nitrogen base collisions with the same particle if there were 10 ppt of the nitrogen base (ammonia or DMA) in the gas phase, which is within the range of possibility.*

13. Page 5, line 32, "to obtain agreement of the chamber" should be "to obtain the agreement of the chamber".

*Changed to "obtain agreement between the chamber particle distribution and the…"*

14. Page 6, line 1, "determine the sizes and total volume" should be "determine the size and the total volume".

*Since several sizes were collected, the use of "sizes" is appropriate here.*

15. Page 6, line 5, "there was" should be "there were".

*The sentence subject is "mode" (singular), and even if this were a collective noun, we are focused on the group, not the individual particles.*

16. Page 12, line 14, "a more stable salt with sulfuric acid than is ammonia", delete "is".

*Changed as requested.*

17. Page 13, line 25, "clusters may is acidic for" should be "clusters may be acidic for"

*Changed as requested.*

18. Page 14, line 20, "(Kim et al., 2016)" this citation should belong to next sentence.

*Changed as requested.*

---

## Author Comment (AC2) · 7 Sep 2016

Response to Anonymous Referee # 2

This is a nicely performed study with very clearly explained measurements. My only suggestions are for revision of the discussion section, where I think the authors may be speculating a bit too much on the implications of their data:

*We thank the reviewer for their generous and frank assessment. To summarize our responses below, we concede that there is not enough evidence to conclude that a phase change occurred over the measured size range. However, we continue to assert that the data demonstrate a size-dependent composition change.*

1. Section beginning with: "The TDCIMS data overall indicate a large increase of base fraction as the particles grew from 10 to 20 nm." I think the authors should be bit more cautious in interpreting their data in this section. Aside from 3 data points, it appears that the base:acid ratios in figure 11 are bound between 0.2 and 0.5 for all measurements. While the results presented are very interesting I think it is premature to conclude that the chemical composition of particles is unquestionably changing as they grow from 10-20 nm.

*In the absence of HTDMA data, there might be reason to doubt the change in particle composition from 10-20 nm. In that case we might state that only sub-13 nm particles ever showed base:acid ratios below about 0.2. However, the HTDMA observations, based on entirely different measurement principles, show the same direction and particle size region for the compositional change.*

*We modified the following sentence (non-italicized part) to emphasize the agreement between instruments:*
"*The direction of this composition change* over this range of diameter *is corroborated by aerosol hygroscopicity results from the same experiments*"

*Based on the Cheng et al. (2015) results, there could also be a phase transition taking place close to this size, and we have added text to that effect, citing Cheng et al.'s results for ammonium sulfate nanoparticles:*

*"At the slightly lower temperature in the present study (273 K), this [phase] transition occurs at a slightly smaller particle diameter, closer to 9 nm. For the dimethylammonium-ammonium-sulfate nanoparticles studied here, a similar size-dependent transition seems plausible."*

*The TDCIMS observations on their own suggest a size-dependent change, and the additional pieces of evidence support this idea. While the possibility of a phase transition may be speculative, the HTDMA observations confirm the size-dependent composition change.*

*Cheng, Y., Su, H., Koop, T., Mikhailov, E. and Pöschl, U.: Size dependence of phase transitions in aerosol nanoparticles, Nat. Commun., 6, 5923, doi:10.1038/ncomms6923, 2015.*

2. Similarly, there is no evidence in the measurements reported here for a phase transition in particles as they grow from 10-20 nm. They do show that 10-20 nm particles sampled are more acidic than basic, and are composed of slightly more ammonia than dimethylamine. More analysis seems necessary to examine the phase of the detected particles.

*The only observational evidence for a phase transition in these experiments is the size-dependent composition change, and we concede that there may be other explanations for this composition change. We argue that since there is a phase change for ammonium sulfate in this size range, it is possible that a similar sulfate salt also has such a transition. The methods used by Cheng et al. to identify the phase state of ammonium sulfate at different sizes can't be applied in the same way in this case, largely because much less is known about the thermodynamics of salts formed from amine-ammonia-sulfuric acid mixtures, and they appear to have properties unique to the individual amines (Chan and Chan, 2013). We have modified the manuscript (described below) to reflect that a phase transition at these sizes is possible but not verifiable using the information currently at hand. We added this discussion to the text:*

*"At the slightly lower temperature in the present study (273 K), this [phase] transition occurs at a slightly smaller particle diameter, closer to 9 nm. For the dimethylammonium-ammonium-sulfate nanoparticles studied here, a similar size-dependent transition seems plausible. However, less is known about the thermodynamics of these mixed ammonium-aminium sulfate salts. Even for large droplets, where size-dependent effects are not important for phase state, pure dimethylaminium sulfate does not form a solid phase, even at < 3 % RH. However, the exposure of such an amorphous liquid droplet to ammonia results in the formation of a solid phase which contains ammonium and aminium and is resistant to further exchange (Chan and Chan, 2013)."*

*We added this statement to the Conclusions as a caveat about the suggested phase change:*

*"However, further work is needed to characterize the phase states of mixed ammonium-aminium-sulfate nanoparticles to assess these possibilities and to understand the growth pathways of such particles."*

*We also removed the statement about a possible phase transition from the Abstract.*

*Chan, L. P. and Chan, C. K.: Role of the Aerosol Phase State in Ammonia/Amines Exchange Reactions, Environ. Sci. Technol., 47, 5755–5762, doi:10.1021/es4004685, 2013.*

Minor Comment:
1. The manuscript refers to "Kim et al 2016". However, this citation does not appear in the references section. Also, I do not believe the "Alhm et al, 2016" citation is published in Aerosol Science and Technology (perhaps it is in review).

*Thanks for pointing out these oversights. We added the Kim et al. reference, and Ahlm et al. has now been published.*